# Adaptation of the Multidimensional Perceived Autonomy Support Scale in Physical Education to Seventh–Tenth Grade Turkish Students: A Validity and Reliability Study

**DOI:** 10.3390/bs15050613

**Published:** 2025-05-01

**Authors:** Mümine Soytürk, Özden Tepeköylü-Öztürk, Henri Tilga

**Affiliations:** 1Department of Physical Education and Sports, Faculty of Sports Sciences, Manisa Celal Bayar University, Manisa 45040, Türkiye; 2Department of Recreation, Faculty of Sport Sciences, Pamukkale University, Denizli 20160, Türkiye; otepekoylu@pau.edu.tr; 3Institute of Sport Sciences and Physiotherapy, Faculty of Medicine, University of Tartu, 51008 Tartu, Estonia; henri.tilga@ut.ee

**Keywords:** self-determination theory, cognitive autonomy support, procedural autonomy support, organizational autonomy support, scale adaptation

## Abstract

Feeling autonomous is a fundamental psychological need for personal growth, integration, development, mental health, and overall well-being. This need is closely linked to the level of support perceived by the environment. This study aimed to analyze the psychometric properties of the Turkish version of the Perceived Multidimensional Autonomy Support in Physical Education Scale (MD-PAS-PE) for Turkish seventh–tenth grade students. The participants comprised a total of 1092 students (606 girls and 486 boys). For obtaining data, the Multi-Dimensional Perceived Autonomy Support Scale for Physical Education, the Basic Psychological Needs Scale in Physical Education, and the Personal Information Form to identify the participants were used. In this study, internal consistency, exploratory factor analysis (EFA), confirmatory factor analysis (CFA), criterion validity, and measurement invariance for gender and age groups statistical analyses were used for data analysis. The extracted factors explained 54.47% of the variance among the 15 items. A fifteen-item, three-factor structure was implemented, as in the original language. In addition, the structural equation model results indicated that cognitive, procedural, and organizational autonomy support sub-dimension scores significantly predicted basic psychological need satisfaction scores. Measurement invariance was examined with gender and age variables, and invariance was observed according to these two factors. It was concluded that the data obtained from the form adapted into Turkish were similar to the original scale, explained by the theoretical structure, and was a valid and reliable scale for measuring autonomy perceptions of seventh–tenth grade students.

## 1. Introduction

Autonomy, which is expressed as all of the capacities necessary for the individual to continue his/her life ([5]), is a broad concept related to the ability of individuals to feel themselves at the center of their actions ([24]) and to manage themselves and make decisions independent of the situation they are in or other individuals. According to [27] ([27]), it represents the internal confirmation that one’s actions originate from oneself. In this context, autonomy significantly affects performance in any situation ([13]). Feeling autonomous is an important basic psychological need for growth, integration, development, mental health and well-being ([4]; [28]). Feeling and behaving autonomously is closely related to the support perceived from the environment ([30]). In this regard, autonomy support—defined as the interpersonal conditions that nurture an individual’s sense of volition ([71])—has become a focal point in contemporary educational research. However, as highlighted in the literature review, although autonomy support has been extensively studied, it is often measured without sufficient attention to the context in which it occurs. This is particularly true in physical education settings, where teacher behaviors and relational dynamics are known to significantly shape how students perceive and internalize autonomy-supportive practices.

The need for autonomy is explained by [25] ([25]) under self-determination theory, which is a general theory of personality and motivation. This theory is a motivation theory based on explaining the psychological processes of individuals’ behavior. According to basic psychological needs theory, one of the sub-theories of self-determination theory, there are three basic needs that are considered universal for all people ([20]; [25]; [74]). These are competence, relatedness, and autonomy. Whether these needs are met or not is closely related to being psychologically healthy, development, and well-being ([7]; [27]; [31]). [74] ([74]) even describe these needs as a fundamental source of psychological nourishment. A complete state of subjective well-being depends on whether basic psychological needs are well met. Competence is related to the need to act effectively and feel competent when using one’s skills and abilities. Relatedness is related to the need to interact with other people and to establish quality relationships in social life ([28]). The need for autonomy, on the other hand, represents individuals’ sense of having willpower and making choices during an action and their inner desire to experience psychological freedom ([24]; [28]). Although these needs are naturally present in human life, whether they are fulfilled depends largely on the degree to which social conditions support healthy individual development ([29]).

Autonomy support within the social context enables individuals to realize their true potential. It also helps them pursue their personal interests and supports the development of intrinsic motivation ([4]; [59]). In this respect, although autonomy seems to be a subjective concept related to the individual’s personality, it is essential that each element, environment, and individuals in the social context provide supportive features. In this context, autonomy support is an interpersonal supportive behavior that enables a person’s internally established voluntary actions, preferences, and values to be included, developed, and nurtured by another person ([71]). Individuals who receive autonomy support can base their behaviors more on internal reasons, their intrinsic motivation increases, they realize their self-potential, and achieve social and psychological satisfaction ([4]; [13]). Autonomy support provided from the social context is met by the family environment, peer relations, and the educational environment. The most effective source of autonomy support in the educational setting is the teacher. In this environment, autonomy support is expressed as helping the learner to realize and develop his/her own goals ([6]). According to [71] ([71]), teachers support students’ autonomy by spending enough time listening to their students, allowing students to speak and work in their own way, using informative praise in feedback, encouraging students to take the initiative and speak, and providing them with clues. Additionally, they foster autonomy by being sensitive to student-generated questions, communicating in a way that expresses understanding of their point of view, asking what students want, providing justifications, and creating different seating arrangements in the classroom. Autonomy-supportive teachers utilize expressions to promote different perspectives and increase their sensitivity to their students’ experiences. Students whose autonomy is supported and motivated in an autonomous way demonstrate a focus on internal reasons and experience a sense of choice and freedom in their actions ([73]). In addition, autonomy support has been deemed to be important for students’ social and emotional well-being as well as their learning performance ([76]). It is stated that students whose autonomy is supported participate more in the educational process, show attention and effort, have positive emotions and make active attempts to positively affect the flow of classroom activities ([72]). When the studies in the literature are examined, it is seen that students who perceive their teachers’ behaviors as autonomy-supportive increase in their learning and motivation for learning ([8]; [32]; [39]; [40]; [62]; [82]) and positively affect and increase their academic achievement ([34]; [62]). Moreover, perceived autonomy support appears to be positively related to students’ various learning behaviors, including the use of deep-level learning strategies and retention during learning ([76]; [88]). Similarly, in physical education, perceived autonomy support from the teacher is associated with basic needs ([54]; [55]), higher level motivational processes such as intrinsic motivation ([14]; [55]), and positive emotions in physical education and leisure time ([21]; [55]). For instance, perceived autonomy support from the physical education environment affects students’ intentions to be physically active in leisure time outside of school ([1]; [42]; [53]). [85] ([85]) found that perceived autonomy support from elements in the context of physical education lessons, including the physical education teacher, played an important role in increasing students’ participation in physical activity as well as their health-related quality of life. Recent evidence highlights the significant role that perceived autonomy support from physical education (PE) teachers plays in shaping students’ physical activity habits and class participation. In a study involving 715 adolescents, teacher autonomy support was found to positively predict students’ physical activity habits while negatively predicting non-participation in PE classes. These findings suggest that supporting students’ autonomy can facilitate both increased participation and the development of a long-term active lifestyle ([48]).

Building upon the theoretical foundations outlined above, the following section synthesizes key findings from the literature and outlines the rationale for measuring students’ perceptions of autonomy support—particularly in the context of physical education.

In summary, the autonomy support perceived by students is related to the learning climate and educational behaviors created by teachers ([7]; [71]). Students who perceive their teachers as autonomy-supportive show improvements in their motivation, learning behaviors, health-related attitudes, and academic outcomes. However, the different individual characteristics of each student suggest that they may perceive autonomy-supportive behaviors from the teacher differently. [83] ([83]) also stated that different students may interpret their teachers’ autonomy support differently, so providing autonomy support only to the student may be insufficient. They emphasized that it is also important to understand how students perceive autonomy support from the teacher source. With this in mind, measuring the perception of autonomy support seems to be important in order to make arrangements for students regarding this parameter. In addition, one of the most important and obvious indicators of whether the behaviors coming from the teacher and arising from the teacher–student relationship are perceived as autonomy-supportive by the student may be the results that can be revealed in the short term through measurement procedures and tools ([18]; [70]).

In this regard, there are different measurement tools utilized in Turkey and Turkish culture. For example, the “Supporting Learner Autonomy Scale” developed by [61] ([61]) is one of these tools. However, this instrument is a scale that makes a general evaluation to determine the extent to which teachers exhibit the behaviors of supporting learner autonomy. In addition, it is a tool where teachers evaluate their own behaviors, not students’ perceptions. Another scale, widely used in this context and adapted to Turkish culture, is the “Perceived Autonomy Support Scale in Exercise” developed by [41] ([41]) and adapted into Turkish by [58] ([58]). This scale addresses autonomy support in a unidimensional structure and is used to measure autonomy-supportive constructs arising not from the physical education context but from the exercise environment in leisure time. A sample item in the form measuring the student’s perception of the teacher’s behavior is as follows: “I think my Physical Education and Sports instructor understands why I prefer to do active sports and/or vigorous exercise in my free time.” As can be seen, the items are related to exercise behaviors in leisure time, not in the physical education environment. Therefore, it is seen that there is a need for a measurement tool that has been adapted to the Turkish culture and handles autonomy support, which is multidimensional due to its structure, in a multidimensional way, and measures the autonomy support arising from the teacher applied to the context of physical education and sports lessons. The study by [49] ([49]) demonstrated that the impact of autonomy support is not uniform; rather, it varies depending on whether the support is cognitive, organizational, or procedural in nature. Therefore, the adaptation of multidimensional measurement tools capable of distinguishing between these types of support is essential for understanding and fostering students’ experiences more effectively. In this scope, the “Multi-Dimensional Perceived Autonomy Support Scale for Physical Education” developed by [83] ([83]) both addresses autonomy support in a multidimensional way and measures the autonomy support arising from the teacher and perceived by the student in the physical education and sports lesson environment. Sample items are as follows: “My Physical Education teacher guides students in finding solutions”, “My Physical Education teacher trusts my ability to succeed in the lesson”. Therefore, the adaptation of this measurement tool into Turkish and its validity and reliability were considered important in order to contribute to both theoretical and practical fields.

In addition to highlighting the scale’s multidimensionality, it is important to clarify its robust theoretical and methodological foundations. The original developers grounded the instrument in self-determination theory ([26], [28]), ensuring that each dimension proposed by [80] ([80])—cognitive, procedural, and organizational autonomy support—reflected a theoretically meaningful facet of autonomy-supportive instruction. They engaged in a systematic, iterative development process, drawing on literature reviews, expert consultations, and pilot testing to establish item clarity and relevance. Rigorous psychometric evaluations, including factor analyses and reliability assessments, further confirmed the measurement’s construct validity and reliability ([83]). By detailing these theoretically grounded and empirically vetted developmental steps, the adapted scale emerges as a more nuanced and robust tool, surpassing the limitations of earlier instruments by offering a comprehensive assessment of autonomy support in physical education contexts.

Measuring perceived autonomy support is crucial because it enables educators and researchers to understand how students interpret and internalize autonomy-supportive behaviors from their teachers, which can significantly impact their motivation, engagement, and academic outcomes ([69]; [74]). By observing and quantifying these perceptions, we can identify gaps between intended teaching practices and students’ actual experiences, allowing for targeted interventions to enhance educational strategies ([6]). Additionally, since cultural factors can influence how autonomy support is perceived, adapting measurement tools to specific cultural contexts ensures that the data collected are accurate and meaningful ([16]). This, in turn, facilitates the development of tailored approaches to foster autonomy-supportive learning environments that promote students’ psychological needs and well-being within their unique cultural settings ([60]).

In developing and adapting measurement tools, it is crucial to ground the process in a strong theoretical framework, ensure thorough methodological rigor, and follow established guidelines for cross-cultural adaptation. The original Multi-Dimensional Perceived Autonomy Support Scale for Physical Education (MD-PASS-PE) was constructed based on self-determination theory ([26]; [74]) ensuring that the cognitive, procedural, and organizational facets of autonomy support reflect well-established dimensions of teacher behavior that foster student motivation and engagement. [83] ([83]) developed the MD-PASS-PE through systematic procedures, such as conducting comprehensive literature reviews on autonomy-supportive teaching, consulting experts in pedagogy and sports psychology for item generation and refinement and performing rigorous psychometric evaluations—including factor analyses and internal consistency tests—to ensure that each item captured a distinct, theoretically meaningful aspect of autonomy support. As a result, the scale advanced beyond previously available unidimensional or context-specific measures by providing a more nuanced tool that recognizes autonomy support as a multifaceted construct. In adapting this measure to Turkish culture, we were attentive to linguistic precision, cultural relevance, and conceptual equivalence across contexts. This entailed iterative translation and back-translation steps, expert panel reviews to confirm content validity, pilot testing with target-age students to gauge clarity and appropriateness of the items, and conducting both exploratory and confirmatory factor analyses to verify the scale’s structural integrity within the new cultural setting. By following these rigorous protocols and highlighting the scale’s multidimensional nature, our adaptation ensures not only that the MD-PASS-PE’s theoretical underpinnings remain intact, but also that the instrument is equally suitable, reliable, and valid for assessing the autonomy support perceived by seventh–tenth grade Turkish students in physical education. This comprehensive approach provides a robust template for future cross-cultural adaptations of autonomy-related measurement tools and strengthens the global utility and interpretability of findings that rely on such multidimensional constructs.

Since the information discussed above is important, this study aimed to analyze the psychometric properties of the Turkish version of the Perceived Multidimensional Autonomy Support in Physical Education Scale (MD-PAS-PE) for Turkish seventh-tenth grade students.

## 2. Materials and Methods

### 2.1. Design of Study

This study employed a quantitative approach, predicated upon a cross-sectional screening model that focuses on statistical analysis to validate the model across diverse student demographics. Such studies examine outcomes across large populations, often investigating differences among them. They are conducted at a single point in time, in contrast to longitudinal studies ([22]; [56]).

### 2.2. Scale Adaptation Process

Adapting a scale from one language and culture to another requires meticulous attention to detail, particularly in ensuring that procedures are followed in the correct sequence. The need for an adapted version of the scale was first evaluated, and it was concluded that its use in the field of physical education and sports within our country would be beneficial.

Following this decision, a translation team was assembled, composed of volunteer experts. In line with [77]’s ([77]) recommendation that at least two experts should be involved in translation processes, we consulted three experts for both the initial translation and the subsequent back-translation phases.

The forward translation of the scale from its original language into Turkish was carried out by three bilingual experts. Two of these individuals had earned their doctoral degrees in the United States, while the third had completed a master’s degree in his home country and had professional coaching experience in the United Kingdom. Their translations were then reviewed collaboratively by the two principal researchers and two additional experts: one, a university lecturer in English with a doctoral degree in Sports Sciences, and the other, an academic with bachelor’s, master’s, and doctoral degrees in English.

Several linguistic refinements were made during this review. For example, the word “understands” in the item “My PE teacher understands my needs” was replaced with the Turkish word “bilir” (“knows”), deemed more culturally and linguistically appropriate. Similarly, the phrase “expressing my opinion” was translated as “to speak my thoughts” to better align with natural Turkish expressions.

Subsequently, the revised Turkish version was reviewed by a high school Turkish language teacher and a university-level Turkish lecturer to ensure linguistic accuracy. Given the scale’s focus on psychological constructs, its content was also examined by a faculty member from the Department of Psychology and a specialist in Sports Psychology. Both confirmed the absence of problematic expressions from a psychological standpoint.

The next step involved back-translation of the Turkish version into English. This task was completed by three additional experts: two of whom had obtained their doctoral degrees in the United Kingdom (in Physical Education and Sports Psychology, respectively), and one who had completed a doctorate in Measurement and Evaluation in the United States. These experts compared the translated items with the original in terms of meaning, grammar, and conceptual equivalence.

In the subsequent Language Validity Testing Phase, two faculty members from the university’s Department of Foreign Languages, each with over a decade of teaching experience, reviewed the scale. Their feedback primarily focused on simplifying item wording to enhance student comprehension. For instance, the item “My PE teacher conveys confidence in my ability to do well in the lesson” was simplified to “My PE teacher believes that I will be successful in class” (translated into Turkish as “Beden eğitimi öğretmenim derste başarılı olacağıma inanır”). The term “talent” was deliberately excluded based on concerns that it might provoke self-doubt in students (e.g., “I am untalented in basketball”). Additionally, the verb “answers” in the item “My PE teacher answers to me when I express my opinion” was replaced with “responds”, which was considered more appropriate in Turkish.

The finalized version of the translated scale was then presented to two practicing physical education and sports teachers—one from a secondary school and the other from a high school—for further feedback. Both affirmed that the items accurately represented real-world teaching behaviors and required no further modification.

An introductory statement was added to the scale to guide respondents during application (e.g., “Please mark the items that you think are most appropriate…”). A pilot study was then conducted with 147 students across two secondary schools and one high school. Students were asked to identify any unclear words or expressions. No comprehension issues or suggestions for revision were reported at this stage. Pilot data analysis resulted in a Cronbach’s alpha value of 0.86. Additionally, confirmatory factor analysis yielded the following results: CMIN/DF = 1.901, RMSEA = 0.079, and CFI = 0.89. Based on these results, it was decided to proceed to the main study. Finally, data collection was completed from the main study, followed by validity and reliability analyses of the Turkish version of the scale (See Appendix A), ([63]; [78]; [91]).

### 2.3. Participants

Students were selected using a convenience sampling technique ([37]), where participants were chosen based on their availability and willingness to participate. This method was appropriate due to the logistical constraints of accessing school-aged students and obtaining necessary permissions. We included schools from diverse socio-economic backgrounds to enhance the representativeness of the sample ([64]). Classes were selected where physical education teachers had been teaching the students for at least two semesters, ensuring sufficient interaction for assessing perceived autonomy support ([90]). In addition, teachers with at least ten years of experience and at least four years at their current school were considered to ensure consistent instructional practices. The demographic characteristics of the participants are shown in Table 1. In addition, the general socioeconomic level of the families was used as a criterion for school selection. Table 2 shows the structure of this evaluation.

The data of 37 participants who did not respond to the items correctly were excluded from the evaluation. The response instructions provided at the beginning of the scales, which were also explained verbally, led to the exclusion of the responses from 16 students who selected multiple options for each item. Additionally, responses were excluded from students who consistently chose only the midpoint option across all items on both scales. In the selection of schools, we considered their socio-cultural backgrounds. We chose schools from high, medium, and low socio-cultural environments and evaluated them as a cluster. When selecting participants’ classes, we preferred the classes of which the current physical education teacher had the longest interaction, with a minimum of two semesters of student-teacher interaction as the lower limit. Students were selected using a convenience sampling technique, and the sample consisted of students who were present at school on the day the scales were administered and who had physical education classes. Although the experiences of the teachers were not considered as a sample in the study, we took into account the data from students of teachers who had at least 10 years of experience and had worked at the current school for at least four years. Teachers with a decade of experience are generally considered to have reached a level of proficiency and stability in their careers ([23]). This criterion helps reduce variability due to differing experience levels and strengthens the reliability of our findings.

Table 1 presents information on the average gender, grade, school, and age of the pilot and main study participants.

In Table 2, schools are categorized into high, medium, and low socio-cultural environments based on the socioeconomic characteristics of the families living in their respective areas. This classification method was selected to provide a more nuanced understanding of the socio-cultural context of each school ([2]).

### 2.4. Procedures

The study received ethical approval from the Manisa Celal Bayar University, Faculty of Medicine Health Sciences Ethics Committee on 2 September 2020 (Approval No. 20.478.486). Additionally, approval was obtained from the Provincial Directorate of National Education on 2 September 2022 (Approval No. E-46949512-605.01-56449208). Data were collected in the second semester of the 2022–2023 academic year, adhering strictly to the regulations set by the National Education authorities, with compliance closely monitored. Informed consent was obtained from all participants and their parents, in accordance with the Declaration of Helsinki. The lead author of the team that developed the scale was contacted and permission was obtained to adapt it to our language and culture. The volunteers who helped us with the translation processes were also taken. In this study, informed consent was obtained from all participants and their parents before data collection and research methods were conducted in accordance with the Declaration of Helsinki.

Due to the structure of the scale to be applied in the research, students and teachers were expected to interact for 7–8 months, even though the necessary official permission was obtained during the COVID-19 period. Since the original scale was developed during the face-to-face education process ([83]), data were not obtained at the stage when the online course was taught, and attendance was not required. In this process, physical education teachers in the selected schools were contacted, and after receiving information that a healthy interaction was achieved, data collection started with the permission of the school principals. Data from the pilot study were obtained from schools outside the main study in May–June 2022. Since there were disruptions in school attendance (absence due to illness, etc.) due to COVID-19, the actual work was delayed for a while. It was postponed until the restrictions were lifted, because in this process, not only was the loss of subjects important, but the most important factor was the healthy and sustainable teacher–student interaction, which came to the forefront in line with the objectives of the research. The main study data were obtained in April, May, and June 2023. All data were collected during physical education classes at schools and in a suitable classroom environment. Before the application of the scales, the students were informed about the purpose of the study, marking the appropriate response level for the scale items, and the anonymity of the information shared, and they were thanked for their participation. The administration of the scales took an average of 20 min.

### 2.5. Measures

In the current study, the Multi-Dimensional Perceived Autonomy Support Scale for Physical Education ([83]), which was sought to be valid and reliable in Turkish, the Basic Psychological Needs Scale in Physical Education ([87]), which was used as a parallel test, and the Personal Information Form to identify the participants, were used.

To provide a theoretically grounded assessment aligned with SDT ([28]), we employed the Basic Psychological Needs in Physical Education Scale as a parallel instrument in this study. SDT posits that the satisfaction of basic psychological needs—autonomy, competence, and relatedness—is essential for fostering intrinsic motivation and psychological well-being. Previous studies have identified autonomy support as a significant predictor of basic psychological needs satisfaction ([68]; [79]), which aligns with the core tenets of SDT. By incorporating this scale, we aimed to assess how autonomy support influences these fundamental needs within the context of physical education, thereby enriching our analysis and ensuring theoretical consistency.

#### 2.5.1. Multi-Dimensional Perceived Autonomy Support Scale for Physical Education (MD-PASS-PE)

Autonomy support of physical education teachers was measured through the Multi-Dimensional Perceived Autonomy Support Scale for Physical Education scale developed by [83] ([83]). This scale consists of three subscales, each consisting of five items. These include cognitive, procedural, and organizational autonomy support. As an example, a sample item for the cognitive sub-dimension of the English scale is “My PE teacher allows me to express my opinion “, for the procedural sub-dimension is “My PE teacher explains the effect of exercises”, and for the organizational sub-dimension is “My PE teacher accepts different solutions in learning of exercises”. The scale is a 7-point Likert-type measurement ranging from 1 = strongly disagree to 7 = strongly agree.

#### 2.5.2. Basic Psychological Needs Scale in Physical Education Scale (BPN-PE)

The BPNES developed by Vlachopoulos and Michailidou (2006) has been modified for Physical Education, and the Basic Psychological Needs in Physical Education scale (BPN-PE) ([89]) was used to measure the extent of participants’ fulfillment of the needs for autonomy, competence, and relatedness in PE. The scale was adapted into Turkish by [87] ([87]). The original scale consists of a total of 12 statements with three factors. After the Turkish form of the scale was applied to 405 secondary school and 412 high school students, the construct validity of the scale was determined by factor analysis, and as a result of the analysis, it was decided that the scale translated into Turkish would consist of 12 statements. During the data collection phase in the study, items were rated on a 5-point Likert scale ranging from 1 = strongly disagree to 5 = strongly agree. Internal consistency analysis in this study showed acceptable values for meeting basic psychological needs (α = 0.82, 95% confidence interval).

#### 2.5.3. Personal Information Form

Through this form, students’ gender, age, grade, and school information were obtained.

### 2.6. Data Analysis

SPSS Statistics and SPSS AMOS (Version 23.0; IBM Corp., Armonk, NY, USA) were used to conduct data analysis. The data were screened for univariate normal distribution based on the [11] ([11]) suggestions; specifically, values for skewness and kurtosis of each item ranging between −7 and +7 are considered as acceptable ([11]). Internal consistency of the scales was evaluated by Cronbach’s alpha coefficients. An exploratory factor analysis (EFA) was carried out in a preliminary evaluation of the factor structure.

The confirmatory factor analysis (CFA) with maximum likelihood (ML) estimation was carried out in a main analysis to evaluate the fit of the proposed factor structure of the scales with the data. Multiple goodness-of-fit indices based on [46] ([46]) were used to examine the model fit: the comparative fit index (CFI), the Bentler–Bonett normed fit index (NFI), the Bentler–Bonett non-normed fit index (NNFI), and the root mean square error of approximation (RMSEA). Fit of the data with the hypothesized model that is considered acceptable is indicated by values ≥ 0.90 for the CFI, NFI and NNFI values, and indicated by values ≤ 0.08 for the RMSEA value. Also, the standardized regression of the items should exhibit weights higher than 0.40 ([43]).

To further examine the dimensionality of the MD-PASS-PE scale and assess the distinctiveness of its subscales, a bi-factor CFA is conducted and several reliability coefficients are calculated based on the factor loadings. Specifically, omega total (ω_t_), omega hierarchical (ω_h_), and omega hierarchical subscale (ω_hs_) coefficients were computed, along with the Explained Common Variance (ECV) and the Value-Added Ratio (VAR) for each subscale.

Measurement invariance across the gender and age groups was examined by multi-group confirmatory factor analysis (MG-CFA). Specifically, the fit values of the first model are compared to the fit values of the second model in which one constraint is added. First, the unconstrained model is proposed, following with more constraints being added in each following step, which assume factor loadings, item intercepts, latent variances, and factor covariances to be equal across the study groups ([11]; [19]; [67]). The values of CFI and the RMSEA were considered for model comparison ([66]). Based on [17] ([17]), invariance is evident when the changes in CFI and RMSEA values are below 0.01 and 0.015, respectively, when a new constraint is added. If the MG-CFA demonstrates that the factor loadings are equal across groups, weak measurement invariance is evident. The strong measurement invariance is evident when factor loadings and item intercepts are equal across groups ([12]).

## 3. Results

In this chapter, explanatory factor analysis, preliminary analysis, factorial validity, analysis of omega coefficients and related metrics, measurement invariance, and criterion validity analyses are presented, which show that the MD-PASS-PE scale adapted into Turkish is a valid and reliable instrument for use in this language.

### 3.1. Exploratory Factor Analysis (EFA)

To further investigate the underlying structure of the MD-PASS-PE, we conducted an EFA prior to the confirmatory factor analyses. The EFA was performed using principal axis factoring with oblique (Promax) rotation to allow for correlations between factors. The Kaiser–Meyer–Olkin measure verified the sampling adequacy for the analysis (KMO = 0.91), and Bartlett’s test of sphericity indicated that correlations between items were sufficiently large for EFA (χ^2^(231) = 5123.45, *p* < 0.001).

The EFA revealed a clear three-factor solution based on the eigenvalues greater than one criterion and the examination of the scree plot. These three factors accounted for 68% of the total variance. All items had factor loadings above 0.50 on their respective factors, with minimal cross-loadings, indicating strong associations between items and their intended factors. The factors corresponded to the proposed dimensions of autonomy support in the MD-PASS-PE: organizational, procedural, and cognitive autonomy support.

These EFA results provided initial empirical support for the three-factor structure of the scale, justifying the subsequent confirmatory factor analyses. The clarity of the factor structure and the strong item loadings suggested that the MD-PASS-PE reliably captures the multidimensional nature of perceived autonomy support in physical education settings.

These results offer strong preliminary evidence supporting the multidimensional structure of the MD-PASS-PE, suggesting that the scale effectively distinguishes between organizational, procedural, and cognitive autonomy support as perceived by students in physical education settings. The clarity and strength of the factor loadings further justify the decision to proceed with confirmatory factor analysis.

### 3.2. Preliminary Analysis

Table 3 presents values of skewness and kurtosis, Cronbach’s alphas, and non-latent correlations among study variables. Univariate normal distribution of the scale items was supported by skewness values ranging from −1.01 to −0.49 and by kurtosis values ranging from 0.76 to 0.82. Cronbach’s alpha values were all acceptable, ranging from 0.76 to 0.82. The non-latent correlations between the study variables range from 0.31 to 0.60. The extracted factors explained 54.47% of the variance among the 15 items.

The preliminary analysis supports the psychometric robustness of the scale, confirming acceptable levels of internal consistency and normality of item distributions. Additionally, moderate-to-strong correlations among the sub-dimensions suggest that, while conceptually distinct, the three types of autonomy support are meaningfully interrelated, reinforcing the appropriateness of a multidimensional framework.

### 3.3. Factorial Validity

To rigorously examine the factor structure of the MD-PASS-PE, a series of confirmatory factor analyses (CFA) were conducted, including models reflecting one-factor, three-factor, and bi-factor solutions. These analyses aimed to validate the multidimensional nature of the scale while testing for the presence of a higher-order general factor.

The expected three-factor structure of the MD-PASS-PE was examined (see Figure 1). Fit values of all the tested models exhibited acceptable model fit values (see Table 4). There were high correlations between the latent autonomy support factors ranging from 0.71 to 0.75. Therefore, the fit of a one-factor model was also examined (see Table 4). Also, the fit indices of the bi-factor model were examined (see Table 4).

To determine the factor structure of the MD-PASS-PE, we conducted CFA testing three models: a three-factor model (Model 1), a one-factor model (Model 2), and a bi-factor model (Model 3). The fit indices for each model are presented in Table 4.

Model 1, the three-factor model, demonstrated acceptable fit indices (χ^2^(87) = 527.392, CFI = 0.920, NFI = 0.906, NNFI = 0.920, RMSEA = 0.068), suggesting that the scale could be represented by three correlated factors of autonomy support. However, the high correlations between the latent factors (ranging from 0.71 to 0.75) indicated substantial overlap among them, raising concerns about discriminant validity.

Model 2, the one-factor model, showed poor fit to the data (χ^2^(90) = 1054.108, CFI = 0.824, NFI = 0.812, NNFI = 0.825, RMSEA = 0.099), implying that a unidimensional structure does not adequately capture the underlying constructs measured by the MD-PASS-PE.

In contrast, Model 3, the bi-factor model, provided the best fit to the data (χ^2^(72) = 268.898, CFI = 0.964, NFI = 0.954, NNFI = 0.964, RMSEA = 0.050), exceeding conventional thresholds for good model fit (CFI and NNFI > 0.95, RMSEA < 0.06). The superior fit of the bi-factor model suggests that while there are specific factors corresponding to different dimensions of autonomy support, there is also a significant general factor influencing all items. This general factor reflects an overarching perception of autonomy support in physical education settings.

Based on these findings, we conclude that the bi-factor model most accurately represents the factor structure of the MD-PASS-PE. The scale effectively captures both the general perception of autonomy support and the specific dimensions within it. This nuanced structure allows for a more comprehensive assessment of autonomy support as experienced by students in physical education, accounting for both shared and unique aspects of the construction.

These findings confirm that while the MD-PASS-PE consists of three distinct sub-dimensions (cognitive, procedural, and organizational autonomy support), the presence of a strong general factor suggests that students also perceive a unified sense of autonomy support in physical education settings. This reinforces the theoretical multidimensionality of autonomy support while validating its overarching structure for holistic evaluation. Thus, both individual subscale scores and a total score may be meaningfully interpreted, depending on the research goal.

### 3.4. Analysis of Omega Coefficients and Related Metrics

To further examine the dimensionality of the MD-PASS-PE scale and assess the distinctiveness of its subscales, we conducted a bi-factor CFA and calculated several reliability coefficients based on the factor loadings. Specifically, we computed omega total (ω_t_), omega hierarchical (ω_h_), and omega hierarchical subscale (ω_hs_) coefficients, along with the Explained Common Variance (ECV) and the Value-Added Ratio (VAR) for each subscale.

The omega total coefficient (ω_t_) for the scale was 0.92, indicating high overall reliability. This suggests that the MD-PASS-PE scale consistently measures perceived autonomy support across its items. Further, the omega hierarchical coefficient (ω_h_), reflecting the proportion of variance attributable to the general factor, was 0.85. This substantial value implies that most of the variance in responses is driven by a common general factor, supporting the scale’s coherence as a unified measure.

For the subscales, the omega hierarchical subscale coefficients (ω_hs_) were the following: cognitive autonomy support (ω_hs_ = 0.24), procedural autonomy support (ω_hs_ = 0.22), and organizational autonomy support (ω_hs_ = 0.20). These relatively low values indicate that while the subscales are conceptually distinct, they contribute little unique variance beyond the general factor. In other words, the added value of scoring each subscale separately is limited.

This interpretation is further supported by the ECV results. The ECV for the general factor was 0.77, while the subfactors had much lower ECV values (0.08 for cognitive, 0.07 for procedural, and 0.08 for organizational autonomy support). The dominance of the general factor’s ECV underscores that the scale is best interpreted as reflecting a single, overarching construct of perceived autonomy support.

Additionally, the VAR values were low across all subscales (0.26 for cognitive, 0.24 for procedural, and 0.23 for organizational autonomy support). Since these are well below the 0.90 threshold, the subscale scores offer minimal additional explanatory power beyond the total score. This finding reiterates that the general score is the most informative and useful indicator for research and practice.

Taken together, these findings suggest that although the MD-PASS-PE scale includes theoretically distinct subscales, in practice, it functions primarily as a unidimensional measure. The general factor accounts for the majority of the variance in responses, making the total score the most robust and meaningful representation of perceived autonomy support in physical education.

### 3.5. Measurement Invariance

To examine the measurement invariance among gender groups, multigroup analyses were conducted to examine the three-factor model in female (n = 606) and male (n = 485) subsamples. Based on the results, the changes in CFI and RMSEA were below 0.01 and 0.015 when the constraints were added to factor loadings and item intercepts.

To examine the measurement invariance among age groups, multigroup analyses were conducted to examine the of the three-factor model among subsamples of seventh graders (n = 295), eighth graders (n = 302), ninth graders (n = 264), and tenth graders (n = 231). Based on the results, the changes in CFI and RMSEA were below 0.01 and 0.015 when the constraints were added to factor loadings and item intercepts. These results demonstrate that students interpret the items of the MD-PASS-PE similarly, regardless of their gender or age group. Thus, the scale is not only psychometrically sound but also practically versatile, allowing researchers and educators to confidently compare autonomy support perceptions across diverse student populations.

### 3.6. Criterion Validity

To further evaluate the external validity of the scale, structural equation modeling was performed using basic psychological needs satisfaction as a criterion variable (see Figure 2). The latent criterion construct was regressed on the latent cognitive autonomy support, organizational autonomy support, and procedural autonomy construct. Based on the results, basic psychological needs satisfaction was significantly predicted by perceived cognitive autonomy support (β = 0.16, *p* = 0.02), perceived organizational autonomy support (β = 0.13, *p* = 0.03), and perceived procedural autonomy support (β = 0.14, *p* = 0.03). In total, the model explained 15% of the variance in the basic psychological needs satisfaction.

## 4. Discussion

### 4.1. Psychometric Modeling

The main object of these study is to analyze the psychometric properties of the Turkish version of the Perceived Multidimensional Autonomy Support in Physical Education Scale (MD-PAS-PE) for Turkish seventh–tenth grade students. The appropriateness of cultural and linguistic adaptation was confirmed by internal consistency, construct, and criterion validity, as well as measurement invariance across gender and age groups. This study is the first adaptation of the scale into Turkish. MD-PASS-PE adapted to different populations ([10]), and those scales have been the subject of many comparative and review studies for assessing children’s perceived autonomy support by physical education teachers ([85]). The analyses conducted in this study showed that the MD-PASS-PE is valid and reliable in measuring the level of autonomy support of seventh–tenth grade students in Turkish society. The translated scale was found to have three sub-dimensions in accordance with the structure of the original scale ([83]), and each item was found to be related to the predicted sub-dimensions. When the bi-factorial model demonstrated superior fit indices, the scale is equally good to measure separate dimensions of autonomy support, as well as autonomy support as a general factor. It was concluded that any analysis based on gender and age is meaningful, may provide a basis for different future studies, and that comparisons based on these two factors show invariance in terms of their outcomes. Based on these results, it can be said that it is an appropriate adaptation for the measurement of the autonomy support perceived by this age group from physical education teachers in Turkish society, and that it is a valid and reliable instrument.

Internal consistency analysis showed that the cognitive, procedural, and organizational autonomy support subscales of the Turkish version of the MD-PASS-PE exhibited acceptable reliability, and each consistently measured a predicted construct (between 0.76 and 0.82; α > 0.70). In the Estonian sample, the internal reliability of the scale ranged between 0.70 and 0.87 ([83]). It ranged between 0.85 and 0.87 in a study with Spanish secondary school students ([10]) and between 0.82 and 0.86 in another study ([86]). In the German version, it ranged between 0.83 and 0.89 ([92]).

The CFA conducted to reveal the structure confirmed the three-factor structure in accordance with the structure of the original scale and showed fit indices (Table 4, Model 1). High correlations of 0.71 to 0.75 were found among the three latent factors. Additional analyses were performed for three different models to see which solutions fit the data better (Table 4). The second examination was compared with a single factor and the expected fit indices were not achieved. This finding showed that cognitive, procedural, and organizational autonomy support sub-dimensions supported the autonomy perception of the scale; in short, the available data for MD-PASS-PE strongly support its three-factor structure. Another analysis was conducted as a bi-factor Model (Table 4, Model 3). The fit indices obtained were found to be at an appropriate level, like Model 1. The two-factor model assumes a general factor on which all items are loaded and a set of orthogonal (unrelated) skill-specific grouping factors. The model is particularly valuable for assessing the empirical persuasiveness of subscales and the practical impact of dimensionality assumptions on test scores ([35]).

The results from the CFA findings align with the three components theorized by [80] ([80]) for autonomy support in the classroom ([80]). The evidence provided serves as empirical support for a theoretical framework, demonstrating that there is a cross-cultural similarity in how physical education teachers approach and support their students. This suggests that regardless of cultural differences, there are commonalities in the ways teachers in this field provide support and guidance to their students. Moreover, the standardized regression weights of the items were all above 0.40 ([43]; [81]). This indicates that each item has a statistically significant relationship with the initially suggested subdimension. In other words, statistical analysis supports the theoretical relationship.

The additional analyses of omega coefficients, ECV, and VAR provide important insights into the dimensionality of the MD-PASS-PE scale. The high omega hierarchical coefficient (ω_h_ = 0.85) and the substantial ECV for the general factor (0.77) indicate that the general factor of perceived autonomy support is the predominant source of variance in the scale scores. The low omega hierarchical subscale coefficients (ω_hs_ ranging from 0.20 to 0.24) and VAR values (ranging from 0.23 to 0.26) for the subscales suggest that they offer limited unique information beyond what is captured by the general factor ([33]; [38]).

These results imply that, although the scale was designed to measure three theoretically distinct dimensions of autonomy support—cognitive, procedural, and organizational—in practice, these dimensions are highly interrelated and may not function as independent constructs within this sample. The strong general factor suggests that students perceive autonomy support from their physical education teachers as a holistic construct rather than distinctly separate dimensions.

From a practical standpoint, this finding suggests that the total score of the MD-PASS-PE scale is a reliable and valid measure of perceived autonomy support in physical education settings. Practitioners and researchers may consider focusing on the total score when assessing autonomy support, as the subscale scores may not provide additional meaningful information.

However, it is important to acknowledge that the theoretical framework underpinning the scale supports the existence of multiple dimensions of autonomy support. The lack of distinctiveness among the subscales in our study could be influenced by cultural factors, the educational context, or the age range of the participants. Further research is needed to explore these possibilities, including studies with diverse populations and settings to determine whether the multidimensional structure holds in different contexts.

Moreover, while the general factor appears to dominate, the subscales may still hold conceptual significance. Teachers and educators might benefit from understanding the different ways autonomy support can manifest, even if these distinctions are not strongly reflected in the statistical analysis. Future studies could investigate interventions targeting specific types of autonomy support to examine their unique effects on student outcomes.

The MD-PASS-PE scale demonstrates strong reliability and validity as a unidimensional measure of perceived autonomy support among Turkish students in grades seven to ten. While the subscales are theoretically meaningful, their practical distinctiveness is limited within this sample. Researchers and practitioners should consider these findings when utilizing the scale and interpreting its results.

Basic psychological needs satisfaction was used as a criterion variable and found to be significantly related to and predicted by the three sub-dimensions of the scale ([15]; [51]). Furthermore, studies have shown that it strongly indicates the variation in meeting basic psychological needs. When adapting the scale for different cultures, the BPN Satisfaction scale was utilized to establish its predictive relationship in terms of criterion validity and the connection between the autonomy support given by physical education teachers and the BPN satisfaction among middle school students in the context of Physical Education ([10]). This association has been explored in both older and more recent studies ([45], [44]; [47]; [52]; [57]; [75]; [84]).

In our research, we found a weak but statistically significant positive relationship between BPN scores and autonomy support scores ([9]; [36]). In a study by Burgueño et al., the relationship between the sub-dimensions of Competence Need Satisfaction and Relatedness Need Satisfaction of the BPN and autonomy support was also found to be weak. However, the relationship was moderate (between r = 0.41 and 0.51) with the Autonomy Need Satisfaction sub-dimension. These parameters align with the theory proposed by Deci and Ryan in 1985b, where one measures the support structure of a need, and the other measures the satisfaction part ([26]).

The original structure of the scale and the consistent results observed in studies conducted across different languages and societies emphasize the common perspective of Turkish society’s teachers on supporting students’ autonomy. In physical education, teachers actively cultivate a positive learning environment by respecting their students’ input and ideas when it comes to carrying out exercises. They encourage open discussions and advocate for the originality of action, a concept referred to as cognitive autonomy support. Moreover, they offer students choices and empower them to have a say in managing the learning processes, known as procedural autonomy support.

Beyond that, teachers allow students to have a voice in selecting exercise methods, making diverse attempts, choosing exercise types, locations, and equipment, and establishing class rules, a practice termed organizational autonomy support. These practices may lead students to feel that their autonomy is supported. To achieve such a result, the teacher should use a student-centered approach in the classroom. Additionally, the teacher’s emphasis on student autonomy may be linked to their demeanor, as someone who values personal autonomy can treat their students with respect and display attitudes and behaviors that promote autonomy in the learning environment.

Many of the teacher’s behaviors in the classroom can be reflective and contribute to students’ growth as individuals who are sharing, respectful, able to solve their problems, work individually and in groups, take responsibility for their behavior and decisions, and participate in classroom decisions. However, it is also crucial to have confidence in the appropriateness of students’ perceptions because autonomy can be confused with violating the desired course structure. In the words of [69] ([69]), the lack of structure in the “let them do what they want!” approach can lead to an overly permissive classroom environment rather than one that supports autonomy ([69]). When evaluated in the context of physical education, this may correspond to the “let them play with the ball” or “give them the ball and let them stall” approach ([3]; [50]; [65]), which may lead to the perception that student autonomy is supported when students are provided the right to play, exercise, speak, and even sit with the equipment they want, the exercises they want, where they want, and with the friends they want, which may be considered quite undesirable in terms of behavioral development.

On the other hand, according to the results of studies conducted in recent years in our country, the most important problems of physical education teachers are reported to be the lack of gymnasium and schoolyard (asphalt/concrete ground) suitable for sports, the lack of importance of the lesson, the attitude of parents and students, the lack of equipment, the lack of knowledge of methods, overcrowded classes, insufficient teaching time, the low development level of students, coping with peer bullying, and undisciplined student profile. The possibility that such problems may be a factor for the teacher to support student autonomy should also be considered.

### 4.2. Academic and Practical Values of MD-PASS-PE Scale

With the adaptation of the scale, it will be possible to determine the approaches of physical education teachers to their students in the Turkish society, which has a traditional culture in its essence, through students’ reactions, to be valid and reliable, and to determine whether our teachers are able to support students’ autonomy at the global level. Identifying the various differences of physical education teachers in our country with its seven regions, wide geography and multicultural structure will provide an opportunity to organize educational and training environments. The variables predicting autonomy that can be treated together with different psychological, social, cognitive, and kinesthetic parameters of the scale can be identified, and it may be possible to focus on these variables in practices. The importance of the weight that should be given to increasing student-centered practices in the renewal phases of teacher education programs for the future will be better understood, and the ground can be prepared for its adoption as the only way to provide superior gains, not just a waste of time. It will provide a preliminary screening opportunity for training on how to support student autonomy through in-service training for physical education teachers, where appropriate.

The results of the scale will show our teachers the level of their approach and what they should pay more attention to in their practice. Supporting students’ autonomy can enhance their academic performance. This improvement can be achieved by involving them in decision-making, problem-solving, independent study, and transferring their knowledge. Another contribution could be to increase the active participation of students, because when they are more involved in their own learning process, they are able to invest more of their attention and energy in the classroom. In this way, they can see and evaluate the emotional and social impact on the student. The student will experience the joy of being an accepted individual in the classroom and of being an individual with boundaries and responsibilities.

## 5. Conclusions

The MD-PASS-PE scale replicated the three-sub-dimensional structure in Turkish with similar results to studies conducted in Estonia, Spain, and Germany. The results showed that the scale is valid and reliable. In addition to being statistically verifiable, this instrument can be used as a feedback tool and can cumulatively determine how teachers’ behavior is perceived based on the information received from students.

Based on the results, the study suggests several promising directions for future research:

The MD-PASS-PE has been adapted into multiple languages, enabling the possibility of multinational studies that facilitate cross-cultural comparisons. This expansion also suggests an opportunity for research designs focused on different geographical regions within Turkey.

Additionally, investigating the long-term effects of autonomy support provided by physical education teachers on students’ academic performance, lifelong physical activity habits, and life skills can help strengthen causal inferences and reveal temporal dynamics.

Research could also explore how autonomy support influences students’ perceptions of autonomy and engagement in a technology-supported virtual or hybrid physical education environment, especially under conditions of mandatory social isolation.

Furthermore, designing experimental studies to test the effectiveness of autonomy support training programs for teachers on student outcomes could be beneficial.

Finally, various effective constructs can be assessed for criterion validity, using scores on motivational orientations, self-regulation, self-image, and similar criteria.

## 6. Limitations and Suggestions

Although the study has many strengths, it also has some limitations. Possible suggestions for future studies within these limitations may be useful to researchers. Official approval for the study was obtained under pandemic conditions, and data collection could not begin for a long period of time. Data could not be collected immediately after the pandemic conditions because we had to wait for the students to have healthy face-to-face interaction with the teacher and classmates. Our main challenge was time. In this case, perhaps the scale could have been tested under such conditions by obtaining online data for teacher autonomy-supportive attitudes in physical education classes conducted online. Then this study could have been repeated with different students.

Another limitation is the location of the city where the study was conducted; this study, which was conducted in western Turkey, may need to be tested in other regions because the conditions may not be similar. The lack of gymnasiums and large class sizes are among the most important problems, so the practices of physical education and the support of teachers’ autonomy may be structurally different in regions with long snow seasons. As emphasized above, if teachers’ approach is “let them play ball” when they do not teach a planned lesson, could students misperceive autonomy support as indifference in the true sense? This question made us researchers think a lot during the study process. Perhaps observation over an extended period could be a reassuring factor for certainty and verification.

The study was conducted with seventh and eighth graders in middle school and ninth–tenth graders in high school. By expanding the population in terms of age group or grade level variables, students’ perceived autonomy support in physical education can be determined in larger populations.

A significant limitation of our study is that the sample was regionally confined to the western part of Turkey. Given Turkey’s vast geographical expanse and substantial socio-economic diversity, this regional focus may limit the generalizability of our findings. The differences in cultural, economic, and educational contexts across the country could influence students’ perceptions of autonomy support in ways not captured by our study. Therefore, while our research provides initial evidence for the validity and reliability of the MD-PASS-PE scale in a Turkish context, future studies should aim to include nationally representative samples. Conducting research across various regions of Turkey would enhance the scale’s applicability and ensure it accurately reflects the diverse experiences of students nationwide. Such efforts would strengthen the evidence base and support the development of interventions tailored to different socio-cultural settings within the country.

A significant limitation of our study is the regional confinement of the sample to the western part of Turkey. Given the country’s vast geographical expanse and considerable socio-economic diversity, this focus may limit the generalizability of our findings to the entire Turkish student population. Practical constraints, including limited time, resources, and the challenges posed by the COVID-19 pandemic, restricted our ability to include a nationally representative sample. Additionally, coordinating data collection across multiple regions would have required substantial logistical efforts beyond our current capacity. Future research should aim to involve participants from various regions and socio-economic backgrounds across Turkey to ensure the scale’s applicability and robustness. Expanding the study in this manner will provide a more comprehensive understanding of perceived autonomy support in diverse educational settings and enhance the validity of the MD-PASS-PE scale nationwide.

Another limitation of this study is the use of a single construct as the criterion variable. While focusing on this primary construct allowed for an in-depth analysis pertinent to the scale under investigation, it may have constrained the comprehensiveness of our validity assessment. Including additional psychological and educational constructs could have provided a more nuanced understanding of the scale’s validity and its predictive capabilities across various outcomes. Future research should consider incorporating a broader range of criterion variables to fully evaluate the relationships between the measured dimensions and broader constructs, thereby enhancing the generalizability and applicability of the scale.

While we recognize that administering the adapted MD-PASS-PE scale to different student populations could have enhanced the robustness of our findings, several constraints prevented us from doing so. Pandemic-related restrictions limited our access to schools and student groups, and the varying timelines for returning to face-to-face instruction made it challenging to coordinate additional data collection within our study period. Consequently, we were unable to test the scale with different students as initially intended. We suggest that future research address this limitation by replicating the study with diverse student samples to strengthen the generalizability and validity of the scale.

## Figures and Tables

**Figure 1 behavsci-15-00613-f001:**
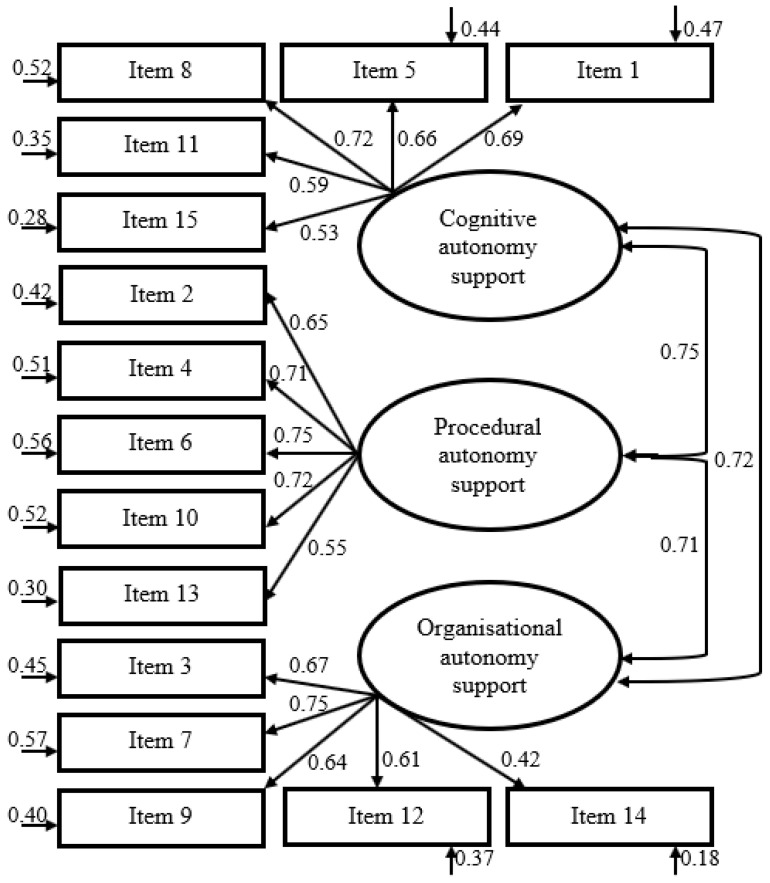
Confirmatory factor analysis of the three-factor model.

**Figure 2 behavsci-15-00613-f002:**
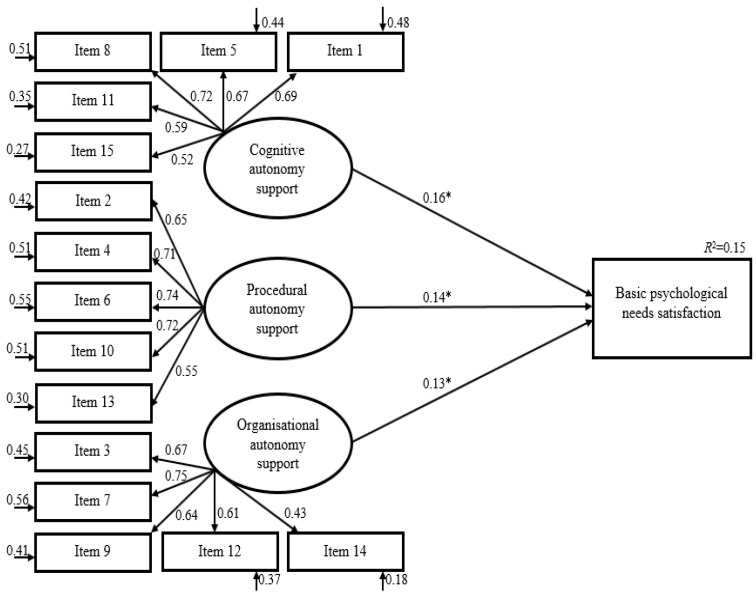
Structural equation modeling predicting basic psychological needs satisfaction in physical education. Note. * *p* < 0.05.

**Table 1 behavsci-15-00613-t001:** Demographic information of pilot and main study participants.

Variables		Main Study		Pilot Study	
		f	%	f	%
Gender	Girl	606	55.5	66	44.9
	Boy	486	44.5	81	55.1
Glass	7th grade	295	27.0	52	35.4
	8th grade	302	27.7	42	28.6
	9th grade	264	24.2	27	18.4
	10th grade	231	21.2	26	17.7
School	1st Secondary School	179	16.4	48	32.7
	2nd Secondary School	222	20.3	53	36.1
	3rd Secondary School	196	17.9	-	-
	1st High School	183	16.8	-	-
	2nd High School	113	10.3	-	-
	3rd High School	199	18.2	46	31.3
Age		M	SD	M	SD
		113.89	1.26	113.99	1.08

**Table 2 behavsci-15-00613-t002:** Classification of schools as socio-cultural environments based on families’ socioeconomic characteristics.

Socioeconomic Level	Settlement Characteristics	Occupational Groups	Income and Employment Stability
Low	Regions predominantly inhabited by families who migrated from Eastern and Central Anatolia	Factory workers, construction laborers, janitorial staff	Low-income levels, often characterized by unstable employment conditions
Medium	Newly developed residential areas primarily occupied by civil servants, public sector employees, and white-collar professionals	Government employees, office workers, public service personnel	Moderately stable income with a diversified occupational landscape
High	Established city center districts and older settlements hosting business owners, tradespeople, and influential community members	Employers, merchants, farmers (both in urban and rural areas)	Higher income levels, with more stable employment opportunities

**Table 3 behavsci-15-00613-t003:** Univariate normal distribution, Cronbach’s alpha coefficients, and correlations between scales.

	M	SD	Skewness	Kurtosis	α	Correlations		
						Cognitive AS	Organizational AS	Procedural AS
Cognitive AS	5.28	1.19	−0.91	0.68	0.77			
Organizational AS	5.08	1.23	−0.61	0.44	0.80	0.55 *		
Procedural AS	5.38	1.24	−1.01	0.80	0.76	0.60 *	0.56 *	
Psychological NS	3.58	0.75	−0.49	3.21	0.82	0.31 *	0.31 *	0.32 *

Note. AS = autonomy support; NS = needs satisfaction; α—Cronbach’s alpha coefficient. * *p* < 0.01.

**Table 4 behavsci-15-00613-t004:** Confirmatory factor analysis results for the study models.

Model			
Model Parameter	Model 1: Three-Factor Model	Model 2: One-Factor Model	Model 3: Bi-Factor Model
χ^2^	527.392	1054.108	268.898
df	87	90	72
CFI	0.920	0.824	0.964
NFI	0.906	0.812	0.954
NNFI	0.920	0.825	0.964
RMSEA	0.068	0.099	0.050

Note. χ^2^ = chi-square; df = degrees of freedom; CFI = comparative fit index; NNFI = Bentler–Bonett non-normed fit index; NFI = Bentler–Bonett normed fit index; RMSEA = root mean square error of approximation.

## Data Availability

The data presented in this study are available on request from the corresponding author due to participants have been informed that their information will not be shared in any way, if persons seeking access to the data explain their reasons and provide appropriate justification.

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
