# Peer review of "Adaptation of the Multidimensional Perceived Autonomy Support Scale in Physical Education to Seventh–Tenth Grade Turkish Students: A Validity and Reliability Study"

_behavsci, 2025, doi:10.3390/bs15050613_

Round 1
Reviewer 1 Report
Comments and Suggestions for Authors
Thank you for the opportunity to review this paper. The process is most rigorous and of value.
I have suggested a series of (very) minor edits to enhance the overall merit. It looks like a lot, however, they connect together to raise the final iteration. I hope you find this of value.
respectfully, Reviewer.

Author Response
Response to Reviewer 1 Comments
General comment of Reviewer 1
Thank you for the opportunity to review this comprehensive paper. Please find notes,
observations, questions, and considerations for collegial discussion and potential action. For ease, I have lettered these. They are each and all quite minor and subtle. Collectively they seek to improve the impact by improving clarity, coherence, and continuity
Kind regards,
Reviewer.
Dear Reviewer 1
Thank you very much for taking the time to review our manuscript. We sincerely appreciate your constructive and insightful feedback. Below, you will find our detailed, point-by-point responses to each of your comments. All corresponding changes have been incorporated into the revised manuscript and are clearly marked using track changes.
Q1: Abstract: Clear and concisely written. It would be useful to know why (the rationale) e.g. in one sentence, as to capture a little of the ‘why’ with the what and how.
Question to consider: Is there any space to do so here?
The dimension of the paper is not present in the abstract. (You may have a reason for this). If pertinent, perhaps discuss and integrate.
R1: We would like to thank the Reviewer for his/her positive comment. We have deleted one sentence due to the very few words (200 words) allowed for the abstract: “Feeling autonomous is a fundamental psychological need for personal growth, integration, development, mental health, and overall well-being. This need is closely linked to the level of support perceived from the environment.” In line with this, we also shaped our purpose statement. In the abstract the dimension of the paper is deleted. These revisions can be seen on lines 15,18
Q2: Keywords: These are well selected as they reflect the breadth and intention of the piece.
R2: Thank you very much for this kind review. Please see the revision on line 37
Q3: Introduction: Autonomy is unpacked, exemplified, and positioned through key and crucial seminal and supporting works, which underpin the current study with the shared theoretical construct of autonomy.
Observation: Articles are mostly beyond the current five-year band (2025-2020), and often the ten-year span. These are all relevant, add value to the links created.
Question: Are there any more current related pieces to refresh the pool of shared workaround a) the concepts if not b) their contextualized application (hence this important study).
Explanations are comprehensive. This is also a desired/required characteristic for an instrument focused piece. This is well done.
Other contributing concepts are illuminated and connected E.g. social and emotional wellbeing, learning (climate, etc) and leisure time- these lead to physical education. The lead into PE is constructed along a pathway, and as such, easy to follow.
R3: Thank you for your kind and thoughtful feedback. We greatly appreciate your observation regarding the temporal scope of the literature used in the introduction. While seminal works were included to establish a strong theoretical foundation, in response to your suggestion, we have updated the literature by incorporating more recent studies (published between 2020 and 2024) focusing on both the concept of autonomy support and its contextualized application in educational and physical education settings. These revisions can be seen on lines as follows:
Koka, A., Tilga, H., Hein, V., Kalajas-Tilga, H., Raudsepp, L. (2021). A Multidimensional approach to perceived teachers’ autonomy support and its relationship with intrinsic motivation of students in physical education. International Journal of Sport Psychology, 52(3), 266-286. doi:10.7352/IJSP.2021.52.266 These revisions can be seen on lines 183-189
Abula, K., Beckmann, J., He, Z., Cheong, C., Lu, F., & Gröpel, P. (2020). Autonomy support in physical education promotes autonomous motivation towards leisure-time physical activity: Evidence from a sample of Chinese college students. Health Promotion International, 35(1), e1–e10. https://doi.org/10.1093/heapro/day102 These revisions can be seen on line 130
Jankauskiene, R., Urmanavicius, D., & Baceviciene, M. (2022). Associations between Perceived Teacher Autonomy Support, Self-Determined Motivation, Physical Activity Habits and Non-Participation in Physical Education in a Sample of Lithuanian Adolescents. Behavioral Sciences, 12(9), 314. https://doi.org/10.3390/bs12090314 These revisions can be seen on lines 134-141
Reeve, J., & Cheon, S. H. (2024). Learning how to become an autonomy-supportive teacher begins with perspective taking: A randomized control trial and model test Teaching and Teacher Education 148 (2024) 104702. These revisions can be seen on lines 159-163
Cheon S., Reeve J.&Marsh H. Autonomy-Supportive Teaching Enhances Prosocial and Reduces Antisocial Behavior via Classroom Climate and Psychological Needs: A Multilevel Randomized Control Intervention. Journal of Sport and Exercise Psychology
These revisions can be seen on lines 159-163
Q4 : Considerations:
Q4 a) Final check for any assertions that remain unsupported- restate or support. E.g. Line
129-132:
“In addition, one of the most important and obvious indicators of whether the behaviors coming from the teacher and arising from the teacher-student relationship are perceived as autonomy-supportive by the student may be the results that can be revealed in the short term through measurement procedures and tools”
Action: could be restated, or supported as is.
R4 a) Dear reviewer, thank you for this careful observation. We have supported our unsupported sentence here with two sources.
Reeve, J., & Cheon, S. H. (2024). Learning how to become an autonomy-supportive teacher begins with perspective taking: A randomized control trial and model test Teaching and Teacher Education 148 (2024) 104702. These revisions can be seen on lines 159-163
Cheon S., Reeve J.& Marsh H. Autonomy-Supportive Teaching Enhances Prosocial and Reduces Antisocial Behavior via Classroom Climate and Psychological Needs: A Multilevel Randomized Control Intervention. Journal of Sport and Exercise Psychology
These revisions can be seen on lines 159-163
Q4 b) Importantly the narrative illuminates and discerns where physical education (PE) is not measured contextually. Some of the rationale is embedded and could be made more explicitin one statement.
Therefore, there appear some lines/parts where the justification emerges. This could be made more explicit at the outset of the piece (introduction).
R4 b) Thank you very much for your valuable feedback regarding the need to clarify the rationale. In response to your comment, stronger and more explicit statements emphasizing the contextual gaps in physical education have been added to the beginning of the manuscript. Please see the revision on lines 52-59 for details.
Q4 c) The paper reads very well. Minor point. Avoid and where able, reposition run on sentences e.g. Line 56: So much so that these needs are expressed as a basic source of nutrition for human psychology by Rayn&Deci (2000) (.e. Keep the sentence balanced and structured around its main clause).
R4 c) Thank you for your positive and kind feedback. Some of the sentences have been revised to ensure clarity and maintain focus on the main clause and core meaning. Should you have additional suggestions regarding other sentences, we would be more than happy to make further revisions. The revised sentences and their corresponding line numbers are provided below.
- “So much so that these needs are expressed as a basic source of nutrition for human psychology by Ryan & Deci (2000).” - “Ryan and Deci (2000) even describe these needs as a fundamental source of psychological nourishment.” Please see the revision on lines 68-70.
- “Although needs exist naturally in human life, one of the important evaluation parameters about whether they are met is whether the social conditions support the healthy development of the individual (Deci & Ryan, 2011).” – “Although these needs are naturally present in human life, whether they are fulfilled depends largely on the degree to which social conditions support healthy individual development (Deci & Ryan, 2011).” Please see the revision on lines 77-79.
- “The autonomy support provided by the social context enables people to reveal their true potential, helps them to reach their subjective interests and desires, and supports their intrinsic motivation (Andersen et al., 2000).” – “Autonomy support within the social context enables individuals to realize their true potential. It also helps them pursue their personal interests and supports the development of intrinsic motivation (Andersen et al., 2000).” Please see the revision on lines 80-82.
- “Students who perceive teacher behavior as supportive of their autonomy are positively affected by their learning, learning-related behaviors, health-related behaviors and academic achievements.” – “Students who perceive their teachers as autonomy-supportive show improvements in their motivation, learning behaviors, health-related attitudes, and academic outcomes.” Please see the revision on lines 145-147.
Q4 d) Observation: The summary for the introduction starts at Line 117 and finishes at 2018.
Question: Is this a summary of what has been shared ahead of Line 117? If so, why does it need such revisiting? If not, then if additional content is being shared, ought there be a piece to bridge the intro, and then the summary?
R4 d) Thank you for this thoughtful observation. The section between Lines 117–218 includes both a synthesis of key points and additional rationale for the current study. To clarify its function, we have revised the section by adding a transition statement and refining the structure to distinguish it from the earlier content. Please see the revision on lines 143-145.
Q4 e) The final statement for Line 2017-219 feels too light and whimsical a statement to accurately reflect the aim and its need, together with the purpose. Consideration Provide an explicit rationale earlier so that this is consolidated in the plenary to this section.
R4 e)
Since the information discussed above is considered important, the aim of this study aimed to analyse the psychometric properties of the Turkish version of the Perceived Multidimensional Autonomy Support in Physical Education Scale (MD-PAS-PE) for Turkish 7th-10th grade students. Please see the revision on line 251-254.
Q5 Materials and Methods
This section is comprehensive. I highlight some observations for consideration. At times, there is a great deal of explanatory information. It could be of value to provide a summary tabulation (rather than just having tables for statistics). (L248-277). For example, the content around the participants in the pilot and main- that could be further conflated for easier consumption, when coupled with a useful table. What is appreciated and crucial to the integrity of the study is the transparency of the approach.
R 5 Thank you for pointing this out. The demographics were tabularized
|
|
|
Main Study |
Pilot Study |
||
|
f |
% |
f |
% |
||
|
Gender |
Girl |
606 |
55.5 |
66 |
44.9 |
|
Boy |
486 |
44.5 |
81 |
55.1 |
|
|
Glass |
7th grade |
295 |
27.0 |
52 |
35.4 |
|
8th grade |
302 |
27.7 |
42 |
28.6 |
|
|
9th grade |
264 |
24.2 |
27 |
18.4 |
|
|
10th grade |
231 |
21.2 |
26 |
17.7 |
|
|
School |
1st Secondary School |
179 |
16.4 |
48 |
32.7 |
|
2nd Secondary School |
222 |
20.3 |
53 |
36.1 |
|
|
3rd Secondary School |
196 |
17.9 |
- |
- |
|
|
1st High School |
183 |
16.8 |
- |
- |
|
|
2nd High School |
113 |
10.3 |
- |
- |
|
|
3rd High School |
199 |
18.2 |
46 |
31.3 |
|
|
|
M |
sd |
M |
sd |
|
|
Age |
13.8 |
1.26 |
13.9 |
1.08 |
|
Please see the revision on lines 354, 356.
Q5 f)
- f) Be consistent with points of view -persons (1st, 2nd, 3rd, …) e.g. line 229 has “… we considered teacher… You have 1st and 3rd- is this intentional? Each read well but at
times the switch is unexpected. What are your thoughts on this? A (your) collective
agreement from a reader's perspective is acceptable.
R5 f) Thank you for this careful advice.
“Additionally, we considered teachers with a minimum of ten years of experience and at least four years at their current school to ensure consistent teaching practices.”
This sentence is to be repleced by:
In addition, teachers with at least ten years of experience and at least four years at their current school were considered to ensure consistent instructional practices. Please see the revision on lines 329-330.
Q5 g) Keep all ethical considerations in one section e.g. L248 “Students participated in the study voluntarily” belongs with ethics section.
Furthermore, one is left wondering- what does this mean and why is this here? If participants are under 18, would there not need to have been a dual process of assent with consent? The ethics section will want to reflect the nature of the context.
Consideration: Sight re-organization to secure the ethical aspects of the context, alongside the simple compliance of institutions and such.
R5 g) Thank you for this suggestion which will save us from too much repetition and misperception. L248 “Students participated in the study voluntarily” deleted.
Participants were under 18 years of age, a dual consent process was used along with assent. Student consent was obtained after parental consent. In this sense, the first sentence in L248 is inappropriate. Please see the revision on line 353.
Q5 h) Observation: L 232- categories of socioeconomic characteristics.
Action: This classification system needs to be supported by a culturally relevant framework. The explanations around this is clear- support is required.
R5 h) The culturally relevant framework to be supported with the following referance;
Akay, E., & Karadag, E. (2019). Multilevel Analyses of Student, Parent, and School Indicators of Achievement in High School Transition in Turkey. School Community Journal, 29(2), 31-62. Please see the revision on line 362.
Q5 i) Insertion of depiction to reduce longer descriptions around demographics.
R5 i) Thank you for pointing this out. The socioeconomic level of school environment was tabularized. Please see the revision on lines 357-359.
|
Socioeconomic Level |
Settlement Characteristics |
Occupational Groups |
Income and Employment Stability |
|
Low |
Regions predominantly inhabited by families who migrated from Eastern and Central Anatolia |
Factory workers, construction laborers, janitorial staff |
Low-income levels, often characterized by unstable employment conditions |
|
Medium |
Newly developed residential areas primarily occupied by civil servants, public sector employees, and white-collar professionals |
Government employees, office workers, public service personnel |
Moderately stable income with a diversified occupational landscape |
|
High |
Established city center districts and older settlements hosting business owners, tradespeople, and influential community members |
Employers, merchants, farmers (both in urban and rural areas) |
Higher income levels, with more stable employment opportunities |
Q5 j) Observation: use of experts to translate. L322. This is well explained. However, this process is not novel. It must be anchored to a floral and research accepted/validated process.
Action- support using the academic literature around this concept/ process and this explicit procedure.
R5 j) You are right, this process is not novel. We added the references in which are explain this process.
Papadakis, N. M., Aletta, F., Kang, J., Oberman, T., Mitchell, A., & Stavroulakis, G. E. (2022). Translation and cross-cultural adaptation methodology for soundscape attributes–A study with independent translation groups from English to Greek. Applied Acoustics, 200, 109031.
Sousa, V.D.; Rojjanasrirat, W. Translation, adaptation and validation of instruments or scales for use in cross-cultural health care research: A clear and user-friendly guideline. J. Eval. Clin. Pract. 2011, 17, 268–274.
WHO. Process of Translation and Adaptation of Instruments. 2023
https://iris.who.int/bitstream/handle/10665/366278/WHO-MSD-GSEDpackage-v1.0-2023.9-eng.pdf
Please see the revision on line 318.
Q5 k) Keep the reporting housed to avoid overlaps, e.g., ethics, as mentioned. Also, see Line367—the pilot study, and now a pretest has been mentioned.
R5 k) You are very right and thank you for your attention, pretest is not appropriate in this sentence, it was intended to talk about the preliminary study to avoid the word pilot study twice.
“To carry out the pilot study, a pre-test was conducted with a total of 147 students in two secondary schools and one high school, and the students were asked to indicate whether there were words and/or expressions they did not understand. At this stage, there were no suggestions.” Please see the revision on line 310.
Q5 l) The order needs to be re-cehced and set- keep that in the temporal order of the organization of the validation journey. The mix ups here serve to mix up the reader.
R5 l) Thank you for your valuable feedback. The order has been revised and the organization has been adjusted according to the temporal order. We believe we have improved clarity and flow for the reader. Please see the revision on line 256.
Q 6) Results
Q6 m) Observation: This is in regard to statistical literacy.
R6 m) So sory. We could not understand exactly what you meant by this “observation” and, therefore, could not find a suitable answer. If you explain it, we will try to answer your “observation.”
Q6 n) The audience is educated and interested a "golden thread" *** from rationale through methods-results and next steps, etc must be created and maintained. Here it is broken.
R6 n) Thank you for your guidance and suggestions. We have been trying to achieve the integrity and fluency you mention; it has been reorganized based on headings according to the flow of time. Please see the revision on line 481.
Q6 o) The reader has to jump into this section. Consider a lead in. e.g. Action: Provide a lead in statement to what these are, what happened. L457
R6 o) An introductory sentence has been added to the conclusion: "In this chapter, explanatory factor analysis, preliminary analysis, factorial validity, analysis of omega coefficients and related metrics, measurement invariance, and criterion validity analyses are presented, which show that the MD-PASS-PE scale adapted into Turkish is a valid and reliable instrument for use in this language. Please see the revision on lines 482-486.
Q6 p) In the analysis the tables are presented with excellent descriptors (what they are). These need to be contextualized interpreted meaningfully for the reader. (I read them I understand what they say.. studied stats and use them)..what do they mean as regards the overall agenda for the work?) Consider a short consolidation statement to each one. e.g. line 513 here is some emerging contextualization "This general factor reflects an overarching perception of autonomy support in physical education settings - this idea wants to be consistently applied.
This action will then enhance and further the impact of L514-519., where some meaning is now emerging.
R6 p) Thank you for this observation, which will increase the clarity of the research results. We added an explanation for this comment in the discussion
Q6 q - Provide connecting bridges across each sub-section. The reporting of what was done is very well executed. To this, stay mindful to illuminate the interpretation e.g. L530-533. Follow that statement with the implication. Bring the reports to life with the consolidation ('so what' of these)
R6 q) We sincerely thank the reviewer for highlighting the importance of establishing stronger interpretative links throughout the results section. In response to comment Q6 q, we have revised each subsection by incorporating brief interpretative statements and connecting transitions to better illuminate the meaning and implications of the findings. We believe these enhancements improve the coherence and narrative flow of the results, and we are grateful for the reviewer’s insightful recommendation. These improvements can be seen in the revised manuscript between lines 508-513, 527-532, 534-538, 563-564, 579-586, 588-593, 639-643, 645-647.
Q7 Discussion
Observation and consideration.
Reads very well. Again the 'what and how are very clear- add the third dimensions of 'why'.
Some pertinent readings are used effectively, and beautifully to link seminal theories- to published works up to more present day. If introducing a new concept (not covered earlier, as in not supported, be sure to support here.
R7 : We are very happy with your words for the discussion section, thank you.
Q7 r) Check e.g. please check whether the theory of constructivism has been anchored to
theory readings ahead of recommendations (L711)- Could have been (my apologies if sojust
a collegial check across this section). The reader needs to know ahead of here, what
these constructs and concepts are in order to then reflect through the discourse presented.
R 7 r) You are right, directly mentioning a concept or theory may have made it difficult to understand. Our sentence has evolved into a more common structure to understand and use. “The teacher should use a constructivist student-centered approach in the classroom to achieve such a result.” Please see the revision on line 787.
Q7 s) Grammatical checks- when read aloud. E.g. L758 invites correction.
R7 s) Thank you for this comment. Grammaticaly checked. Please see the revision on line 853-856
“The scale results will show our teachers the level of their approach and what they should pay more attention to in their practice.
Supporting students' autonomy can enhance their academic performance. This improvement can be achieved by involving them in decision-making, problem-solving, independent study, and transferring their learning.” Please see the revision on lines 835-838.
Q8 Conclusions: Limitations are authentic and strengthen the merit of what was illuminated. Very limited.
Consideration: Can more attention be shared to this section? Limitations (and suggestions') -check the journal for the phrase thank you- see previous articles.
Rigorous overview.
R8: We want to express our gratitude for the valuable information you have provided. We have added a thank you section, though it does not adhere to the Journal's usual format. We want to express our gratitude to the physical education teachers who supported us in reaching a wide audience, facilitated the process, and made efforts to contact parents to obtain permission to obtain data. There's no harm in taking it off, of course.
There's no harm in taking it off, of course.
Please see the revision on lines 852- 870, 963
Dear reviewer, thank you very much for your observations, criticism and comment.
Please see attachment.

Reviewer 2 Report
Comments and Suggestions for Authors
This paper has been very interesting to read overall. While admittedly the authors might not speak English as their first language, they have done particularly well to organise and explain their study at a standard of English which is commendable. The literature review has been carefully examined and presented although I would like to ask if the most current literature has been sourced. Given that the approvals process and data collection were disrupted by the COVID-19 pandemic, my encouragement would be for the authors to keep some of the seminal work published by well-known authors on autonomy and self-determination theory (e.g., the early work done by Deci & Ryan, de Charms), [unless there is anything more current to include from these authors] and to look for some work published within the past five years in these areas. There are some additional, specific comments in a separate document which hopefully will be useful to the authors in the redraft. Well done.

Author Response
Dear Reviewer 2
We would like to express our sincere gratitude for your review of our manuscript. Your comments have been highly valuable in refining the quality and clarity of our work. Below, we provide our detailed, point-by-point responses to each of your suggestions. All corresponding revisions have been incorporated and are indicated in the manuscript using track changes.
Q1 General Comments
This paper has been very interesting to read overall. While admittedly the authors might not speak English as their first language, they have done particularly well to organise and explain their study at a standard of English which is commendable. The literature review has been carefully examined and presented although I would like to ask if the most current literature has been sourced. Given that the approvals process and data collection were disrupted by the COVID-19 pandemic, my encouragement would be for the authors to keep some of the seminal work published by well-known authors on autonomy and self-determination theory (e.g., the early work done by Deci & Ryan, de Charms), [unless there is anything more current to include from these authors] and to look for some work published within the past five years in these areas.
R1: Thank you for your words of praise for the language of the article and the effort we put into it.
The reason why the references we use are dated is due to our desire to present the theoretical framework well and to reflect the main structure of the framework. Some references in the literature are of “golden” value that do not lose their value. Nevertheless, it would be nice with new sources, we tried to add them. Please see the revision on lines 137, 159, 179.
Q2 Abstract
The abstract has been written well, with attention being given to conciseness and accuracy.
R2: Thank you for your attention to the appropriate quality of our summary. We have tried to express it in 200 words.
Q2 (i) Line 17: “The participants comprised a total…”
R2 (i) : Thank you for this correction. Please see the revision on line 20.
Q2 (ii) Line 30: “It was concluded…”
R2 (ii) : Thank you for this correction. Please see the revision on line 33.
Q3 Introduction
Q3 (i) Line 48: “…Deci and Ryan (1985)…” [the ampersand is not to be used outside of parentheses, so please check the remainder of the manuscript for any instances of this and correct them]
Q3 (i) Thank you, the corrections requested by you have been made.
Q3 (ii) There are several instances in Para. 2 p. 2 where double quotation marks have been used, but there is no direct quotation included (which would be accompanied by a page number). Please scan the manuscript carefully and adjust any instances of double quotation marks accordingly (this will probably also be done at the copy-editing stage, if the paper is accepted for publication).
R3 (ii) Thank you for your attentive observation regarding the use of double quotation marks. We have carefully reviewed the manuscript and removed all unnecessary quotation marks that were not associated with direct citations. In cases where direct quotations were used, the appropriate page numbers have now been included. These corrections can be found on page 2, paragraph 2, and throughout the manuscript where relevant. Please see the revision on lines 60, 63
Q3 (iii) Line 55: “…healthy, development and well-being…”
R3 (iii) Thanks for your review, the typo has been corrected. Please see the revision on line 67.
Q3 (iv) Line 78: “…self-potential…”
R3 (iv) We sincerely appreciate your input. I've made the necessary changes based on your suggestion. Please see the revision on line 90.
Q3 (v) pp. 2-3: There is a rather lengthy paragraph where it is suggested the content is spread evenly across two shorter paragraphs for enhanced reader impact.
R3 (v) It is really a very long sentence, thank you for your observation. Changed with:
“According to Reeve and Jang (2006), teachers support students' autonomy by spending enough time listening to their students, allowing students to speak and work in their own way, using informative praise in feedback, encouraging students to take the initiative and speak, and providing them with clues. Additionally, they foster autonomy by being sensitive to student-generated questions, communicating in a way that expresses understanding of their point of view, asking what students want, providing justifications, and creating different seating arrangements in the classroom.” Please see the revision on lines 95-103.
Q4 Materials and Methods
This section has been drafted well, with careful attention given to describing how the original scales were developed from English into Turkish.
R4: Thank you for your meticulous evaluation of this chapter. We have addressed and edited each of your suggestions, and we appreciate your help with language editing and ensuring that our work is perceived correctly.
Q4 (i) Line 296: “…data were not…” and Line 307: “…main study data were…”
R4 (i) Thanks; edited according to your suggestion. Please see the revision on lines 380, 391
Q4 (ii) Line 329: “…and doctoral degree…”
R4 (ii) Thanks, edited according to your suggestion. Please see the revision on line 272.
Q4 (iii) Lines 322-371: This paragraph is especially long, please divide the content among two shorter paragraphs.
R4 (iii): Thank you for this valuable review. We have tried to structure this section more clearly by breaking it into smaller sections. Please see the revision on lines 257-318.
Q4 (iv) Line 421: “…data were screened…”
R 4 (iv): Thanks, edited according to your suggestion. Please see the revision on line 445.
Q4 (v) Lines 441-442: “…computed…” or “…calculated…”
R4 (v) Thanks, edited according to your suggestion. Please see the revision on line 465.
Q5 Results and Discussion
R5 Thank you for your comments in this section.
Q5 (i) Lines 699-738: The length of the paragraph needs to be addressed.
R5 (i) Please see the revision on lines 774-815.
The original structure of the scale and the consistent results observed in studies conducted across different languages and societies emphasize the common perspective of Turkish society's teachers on supporting students' autonomy. In physical education, teachers actively cultivate a positive learning environment by respecting their students' input and ideas when carrying out exercises. They encourage open discussions and advocate for the originality of action, a concept referred to as cognitive autonomy support. Moreover, they offer students choices and empower them to have a say in managing the learning processes, known as procedural autonomy support.
Beyond that, teachers allow students to have a voice in selecting exercise methods, making diverse attempts, choosing exercise types, locations, and equipment, and establishing class rules, a practice termed organizational autonomy support. These practices may lead students to feel that their autonomy is supported. To achieve such a result, the teacher should use a student-centered approach in the classroom. Additionally, the teacher's emphasis on student autonomy may be linked to their demeanor, as someone who values personal autonomy can treat their students with respect and display attitudes and behaviours that promote autonomy in the learning environment.
Many of the teacher's behaviours in the classroom can be reflective and contribute to their growth as individuals who are sharing, respectful, able to solve their problems, able to work individually and in groups, able to take responsibility for their behaviour and decisions, able to participate in classroom decisions... in the classroom environment. In addition, it is also crucial to have confidence in the appropriateness of students' perceptions because autonomy can be confused with violating the desired course structure. In the words of Reeve (2006), the lack of structure in the "let them do what they want!" approach can lead to an overly permissive classroom environment rather than one that supports autonomy (Reeve, 2006) .
When evaluated in the context of physical education, this may correspond to the "let them play with the ball" or "give them the ball and let them stall" approach (Alagül & Gürsel, 2019; Koşar, 2007; Pehlevan et al., 2019) . Which may lead to the perception that student autonomy is supported when students are provided the right to play, exercise, speak, and even sit with the equipment they want, the exercises they want, where they want, and with the friends they want, which may be considered quite undesirable in terms of behavioural development. On the other hand, according to the results of studies conducted in recent years in our country, the most important problems of physical education teachers are reported to be lack of gymnasium and schoolyard (asphalt/concrete ground) not suitable for sports, lack of importance of the lesson, attitude of parents and students, lack of equipment, lack of knowledge of methods, overcrowded classes, insufficient teaching time, low development level of students, coping with peer bullying and undisciplined student profile. The possibility that such problems may be a factor for the teacher to support student autonomy should also be considered.
Q6 Conclusions, Limitations and Suggestions
These two sections have been drafted particularly well.
R6 Thank you very much for your kind and encouraging feedback. We truly appreciate your recognition of the effort put into drafting these sections and are pleased to know that they were found to be well-structured and clear.
Q6 (i) Line 790: “…large class sizes…”
R6 (i) In response to the reviewer’s observation, the expression “high class sizes” has been revised to “large class sizes” to ensure clarity and precision in terminology. The revisions can be seen on line 886..
Q 7 References
The references have been drafted in a particularly impressive way, using the conventions of APA7 guidelines accurately. However, and as noted in the General Comments section, out of the 70 references cited, only 12 have been published in the past five years (or since 2020). My strong encouragement would be for the authors to improve this ‘proportion of current literature’ significantly.
R7 Thank you very much for your valuable observation. We appreciate your note regarding the proportion of recent literature. In response, we have thoroughly reviewed the reference list and identified opportunities to integrate more current sources. Accordingly, we have added several studies published within the last five years to strengthen the theoretical foundation and enhance the relevance of our discussion. These updates can be observed in the revised manuscript, particularly in the Introduction and Discussion sections (Abula et al., 2020; Brisimis et al., 2020; Charlot Colomès et al., 2021; Cheon et al., 2022, 2023; Ebersold et al., 2019; Giudice, 2024; Jankauskiene et al., 2022; Koka et al., 2021; Langøy et al., 2024; Papadakis et al., 2022; Reeve & Cheon, 2024; WHO (World Health Organization), 2023)We are grateful for your recommendation, which has helped us improve the currency and impact of the manuscript.
List of added references
Abula, K., Beckmann, J., He, Z., Cheong, C., Lu, F., & Gröpel, P. (2020). Autonomy support in physical education promotes autonomous motivation towards leisure-time physical activity: evidence from a sample of Chinese college students. Health Promotion International, 35(1), 1–10. https://doi.org/10.2307/48554469
Brisimis, E., Krommidas, C., Galanis, E., Karamitrou, A., Syrmpas, I., & Comoutos, N. (2020). Exploring the Relationships of Autonomy- supportive Climate, Psychological Need Satisfaction and Thwarting with Students’ Self-talk in Physical Education. Journal of Education, Society and Behavioural Science, 112–122. https://doi.org/10.9734/jesbs/2020/v33i1130276
Charlot Colomès, A. A., Duchesne, S., & Boisclair Châteauvert, G. (2021). Autonomy support and school adjustment: The mediating role of basic psychological needs. International Journal of School and Educational Psychology, 9(sup1), S182–S200. https://doi.org/10.1080/21683603.2021.1877226
Cheon, S. H., Reeve, J., & Marsh, H. W. (2023). Autonomy-Supportive Teaching Enhances Prosocial and Reduces Antisocial Behavior via Classroom Climate and Psychological Needs: A Multilevel Randomized Control Intervention. Journal of Sport and Exercise Psychology, 45(1), 26–40. https://doi.org/10.1123/jsep.2021-0337
Cheon, S. H., Reeve, J., Marsh, H. W., & Song, Y. G. (2022). Intervention-enabled autonomy-supportive teaching improves the PE classroom climate to reduce antisocial behavior. Psychology of Sport and Exercise, 60. https://doi.org/10.1016/j.psychsport.2022.102174
Ebersold, S., Rahm, T., & Heise, E. (2019). Autonomy support and well-being in teachers: differential mediations through basic psychological need satisfaction and frustration. Social Psychology of Education, 22(4), 921–942. https://doi.org/10.1007/s11218-019-09499-1
Giudice, M. Del. (2024). g-Disattenuation: Using Omega Coefficients to Estimate Effect Sizes for the Underlying General Factors. https://doi.org/https://doi.org/10.31234/osf.io/zutrx
Jankauskiene, R., Urmanavicius, D., & Baceviciene, M. (2022). Associations between Perceived Teacher Autonomy Support, Self-Determined Motivation, Physical Activity Habits and Non-Participation in Physical Education in a Sample of Lithuanian Adolescents. Behavioral Sciences, 12(9). https://doi.org/10.3390/bs12090314
Koka, A., Tilga, H., Hein, V., Kalajas-Tilga, H., & Raudsepp, L. (2021). A Multidimensional approach to perceived teachers’ autonomy support and its relationship with intrinsic motivation of students in physical education. International Journal of Sport Psychology, 52(3), 266–286. https://doi.org/10.7352/IJSP.2021.52.266
Langøy, A., Diseth, Å., Wold, B., & Haug, E. (2024). Autonomy support, basic needs satisfaction, and involvement in physical education among Norwegian secondary school students. Frontiers in Psychology, 15. https://doi.org/10.3389/fpsyg.2024.1505710
Papadakis, N. M., Aletta, F., Kang, J., Oberman, T., Mitchell, A., & Stavroulakis, G. E. (2022). Translation and cross-cultural adaptation methodology for soundscape attributes – A study with independent translation groups from English to Greek. Applied Acoustics, 200. https://doi.org/10.1016/j.apacoust.2022.109031
Reeve, J., & Cheon, S. H. (2024). Learning how to become an autonomy-supportive teacher begins with perspective taking: A randomized control trial and model test. Teaching and Teacher Education, 148. https://doi.org/10.1016/j.tate.2024.104702
WHO (World Health Organization). (2023). Adaptation and translation guide. https://iris.who.int/bitstream/handle/10665/366278/WHO-MSD-GSEDpackage-v1.0-2023.9-eng.pdf
Some general comments which need to be addressed:
Q7 (i) Various references include capital letters in the titles [e.g., Byrne, M.B. (2010)].
R7 (i) This name order error has been fixed. 996-997
Q7 (ii) Various book references where an edition number is used [e.g., Byrne, M.B. (2010), Hair et al. (2019)] need to have the edition as an ordinal number, e.g., (7th ed.) not in words (Seventh edition).
R7 (ii) Errors in this aspect have been fixed.
Dear reviewer, thank you very much for your observations, criticism and comment.
Please see attachment.

Reviewer 3 Report
Comments and Suggestions for Authors
First of all, I would like to congratulate you on obtaining such a large sample for your study. I would like to give you a number of comments to improve your work.
Line 157 a phrase in inverted commas appears in italics and the next one does not. Revise to establish the same criteria.
Line 219 incorporate the stated hypothesis into the study objective.
Lines 261 to 263, you indicate that you excluded questionnaires that were incorrectly completed. Could you please clarify what criteria were used to eliminate them?
Lines 400 and 401 are results. I suggest that you do not mix paragraphs and incorporate these sentences in the corresponding section.
Lines 507 to 519: the main problem I see in the paper is that you indicate that Model 3 is the best fit to the data, but yet the whole paper is developed with model 1. I suggest to thoroughly revise this section and the rest of the paper as it would change the discussion and conclusions.
Author Response
Dear Reviewer 3
Thank you for your careful and thoughtful review of our manuscript. We truly appreciate the time and effort you dedicated to providing such insightful feedback. We have addressed each of your comments in the responses below, and corresponding changes have been made throughout the revised manuscript, highlighted using track changes.
Q1 General Comment
First of all, I would like to congratulate you on obtaining such a large sample for your study. I would like to give you a number of comments to improve your work.
R1: Thank you very much for this comment.
Q2: Line 157 a phrase in inverted commas appears in italics and the next one does not. Revise to establish the same criteria.
R2 : "My Physical Education teacher trusts my ability to succeed in the lesson". Please see the revision on the lines 189-190.
Q3 Line 219 incorporate the stated hypothesis into the study objective.
R3: Thank you for this useful review. We thought our shortcoming was that we presented the statements incompletely. We have edited them in accordance with the suggestions of the other referees and your suggestion. Please see the revision on lines 251-254.
Q4: Lines 261 to 263, you indicate that you excluded questionnaires that were incorrectly completed. Could you please clarify what criteria were used to eliminate them?
R4: Dear reviewer, we could not decide the correct way to mark more than one item.
On the other hand, middle item actually means undecided and it was thought that participants who responded undecided for all items would disrupt the cleanliness of the data. Please see the revision on lines 335-339.
Q5: Lines 400 and 401 are results. I suggest that you do not mix paragraphs and incorporate these sentences in the corresponding section.
R5: This sentence was deleated: “In the version adapted to Turkish, as a result of Confirmatory Factor Analysis, a structure with three sub-factors was obtained similar to the original.”
Q6: Lines 507 to 519: the main problem I see in the paper is that you indicate that Model 3 is the best fit to the data, but yet the whole paper is developed with model 1. I suggest to thoroughly revise this section and the rest of the paper as it would change the discussion and conclusions.
R6: Thank you for this observation, which will increase the clarity of the research results. We added an explanation for this comment in the discussion. Please see the revision on lines 673, 700, 724-734.
Dear reviewer, thank you very much for your observations, criticism and comment.
Please see attachment.

Reviewer 4 Report
Comments and Suggestions for Authors
Article: Adaptation of the Multidimensional Perceived Autonomy Support Scale in Physical Education to 7-11 Grades in Turkish: A Validity and Reliability Study
Areas for Improvement:
- The abstract recommends including a concise description of the methodology used (type of study, population, instruments, and analysis). This is important for understanding the scope and rigor of the work.
- In the introduction, the authors describe self-determination. It is recommended that a bibliographic review be conducted over the last 10 years, as documents dating back 30 years have been included (Deci & Ryan, 1985). Likewise, the study's objective should be reviewed and should be formulated coherently and included in all required sections, such as the abstract.
- In the methodology section, authors are encouraged to develop this section in a sequence. This could include a description of the research design and type, population and sample (inclusion and exclusion criteria), techniques and instruments, data collection procedures, and type of analysis. Additionally, explain how a pilot study was conducted and its implications for the main study.
- In the results section, it would be important to provide a better description of the results obtained to allow the reader to better understand them.
- We suggest improving the wording of the results, providing greater detail and clarity in the presentation of each table, if necessary, to facilitate reader understanding.
- In the discussion section, we recommend integrating recent studies relevant to the study's object of study. This will allow the study's findings to be compared with the current literature and reinforce the scientific relevance of the results.
- In the conclusion, we suggest incorporating a subsection on new lines of research in this section.
- It is suggested that the validated instrument be included as an annex to the article.
Author Response
Dear Reviewer 4
We are grateful for your constructive feedback and the time you invested in reviewing our manuscript. Your thoughtful observations have helped us strengthen the paper. Please find below our point-by-point responses to your comments. All revisions have been carefully implemented and are clearly marked in the updated version of the manuscript.
Q1: The abstract recommends including a concise description of the methodology used (type of study, population, instruments, and analysis). This is important for understanding the scope and rigor of the work.
R1: Thank you for your valuable review. Nevertheless, we thought that the information in the title could be expressed by the fact that it is a scale adaptation - screening, the answer to the question of who - age group, and the instruments - two measurement tools in the abstract. Please see the revision on line 15.
Q2: In the introduction, the authors describe self-determination. It is recommended that a bibliographic review be conducted over the last 10 years, as documents dating back 30 years have been included (Deci & Ryan, 1985). Likewise, the study's objective should be reviewed and should be formulated coherently and included in all required sections, such as the abstract.
R2: Thank you for this thoughtful detailed observation. We have revised our purpose statement and made it more emphatic and descriptive. Please see the revision on lines 18, 251.
In addition, we greatly appreciate your observation regarding the temporal scope of the literature used. While seminal works were included to establish a strong theoretical foundation, in response to your suggestion, we have updated the literature by incorporating more recent studies (published between 2020 and 2024) focusing on both the concept of autonomy support and its contextualized application in educational and physical education settings.
Koka, A., Tilga, H., Hein, V., Kalajas-Tilga, H., Raudsepp, L. (2021). A Multidimensional approach to perceived teachers’ autonomy support and its relationship with intrinsic motivation of students in physical education. International Journal of Sport Psychology, 52(3), 266-286. doi:10.7352/IJSP.2021.52.266 These revisions can be seen on lines 183-189
Abula, K., Beckmann, J., He, Z., Cheong, C., Lu, F., & Gröpel, P. (2020). Autonomy support in physical education promotes autonomous motivation towards leisure-time physical activity: Evidence from a sample of Chinese college students. Health Promotion International, 35(1), e1–e10. https://doi.org/10.1093/heapro/day102 These revisions can be seen on line 130
Jankauskiene, R., Urmanavicius, D., & Baceviciene, M. (2022). Associations between Perceived Teacher Autonomy Support, Self-Determined Motivation, Physical Activity Habits and Non-Participation in Physical Education in a Sample of Lithuanian Adolescents. Behavioral Sciences, 12(9), 314. https://doi.org/10.3390/bs12090314 These revisions can be seen on lines 134-141
Reeve, J., & Cheon, S. H. (2024). Learning how to become an autonomy-supportive teacher begins with perspective taking: A randomized control trial and model test Teaching and Teacher Education 148 (2024) 104702. These revisions can be seen on lines 159-163
Cheon S., Reeve J.&Marsh H. Autonomy-Supportive Teaching Enhances Prosocial and Reduces Antisocial Behavior via Classroom Climate and Psychological Needs: A Multilevel Randomized Control Intervention. Journal of Sport and Exercise Psychology
These revisions can be seen on lines 159-163
Q3: In the methodology section, authors are encouraged to develop this section in a sequence. This could include a description of the research design and type, population and sample (inclusion and exclusion criteria), techniques and instruments, data collection procedures, and type of analysis. Additionally, explain how a pilot study was conducted and its implications for the main study.
R3: Thank you for your suggestions, we thought we had implicitly reflected the inclusion and exclusion criteria. Inclusion criteria: willingness to participate, age and grade level, compulsory and active participation in physical education (students on sick leave were excluded). Exclusion criteria: students who did not want to participate or whose parents did not allow them to participate, and those who responded inappropriately to the scale items (more than one response to an item or all "undecided" responses).
The pilot study involved 147 students. Data analysis resulted in a Cronbach's alpha value of 0.83.
Additionally, confirmatory factor analysis yielded the following results: CMIN/DF = 1.901, RMSEA = 0.079, and CFI = 0.89. The assessment of normality produced a c.r. value of 23.162, which was found to be relatively close to normality; we suspect that the sample size, being less than 200, may have influenced this outcome.
Moreover, the loadings for the 11th and 15th items were below 0.40. Although these items were not removed from the scale, we considered whether it would be necessary to address this in the main data analysis, again noting that the smaller sample size might be a contributing factor. These results are presented in lines: 313-318.
N=147
CMIN/DF = 1,901
RMSEA = ,079
CFI= ,89
Q4: In the results section, it would be important to provide a better description of the results obtained to allow the reader to better understand them.
R4: Thank you for this suggestion. Edits have been made in the results section in accordance with your suggestion. Emphasis has been placed on making the results more understandable in simpler language.
Q5: We suggest improving the wording of the results, providing greater detail and clarity in the presentation of each table, if necessary, to facilitate reader understanding.
R5: Thank you for that suggestion, we have done that for every table. We hope these explanations will help the reader to read more easily.
Please see the revision on lines: 508, 527, 580.
Q6: In the discussion section, we recommend integrating recent studies relevant to the study's object of study. This will allow the study's findings to be compared with the current literature and reinforce the scientific relevance of the results.
R6: In accordance with your valuable suggestion in the discussion section we added recent studies relevant to the study's object of study. Please see the revision on lines 717, 766.
Giudice, M. D. (2024). g-Disattenuation: Using Omega Coefficients to Estimate Effect Sizes for the Underlying General Factors. https://doi.org/10.31234/osf.io/zutrx
Deng, L., & Chan, W. (2017). Testing the Difference Between Reliability Coefficients Alpha and Omega. Educational and Psychological Measurement, 77(2), 185-203. https://doi.org/10.1177/0013164416658325
Charlot Colomès, A. A., Duchesne, S., & Boisclair Châteauvert, G. (2021). Autonomy support and school adjustment: The mediating role of basic psychological needs. International Journal of School & Educational Psychology, 9(sup1), S182-S200.
Langøy A, Diseth Å, Wold B and Haug E (2024) Autonomy support, basic needs satisfaction, and involvement in physical education among Norwegian secondary school students. Front. Psychol. 15:1505710. doi: 10.3389/fpsyg.2024.1505710
Ebersold, S., Rahm, T., & Heise, E. (2019). Autonomy support and well-being in teachers: differential mediations through basic psychological need satisfaction and frustration. Social Psychology of Education, 22(4), 921–942. https://doi.org/10.1007/S11218-019-09499-1
Brisimis, E., Krommidas, C., Galanis, E., Karamitrou, A., Syrmpas, I., & Comoutos, N. (2020). Exploring the Relationships of Autonomy- supportive Climate, Psychological Need Satisfaction and Thwarting with Students’ Self-talk in Physical Education. 112–122. https://doi.org/10.9734/JESBS/2020/V33I1130276
Q7: In the conclusion, we suggest incorporating a subsection on new lines of research in this section.
R7: Based on the results, the study suggests several promising directions for future research. Please see the revision on lines 852-870
The MD-PASS-PE has been adapted into multiple languages, enabling the possibility of multinational studies that facilitate cross-cultural comparisons. This expansion also suggests an opportunity for research designs focused on different geographical regions within Turkey.
Additionally, investigating the long-term effects of autonomy support provided by physical education teachers on students' academic performance, lifelong physical activity habits, and life skills can help strengthen causal inferences and reveal temporal dynamics.
Research could also explore how autonomy support influences students' perceptions of autonomy and engagement in a technology-supported virtual or hybrid physical education environment, especially under conditions of mandatory social isolation.
Furthermore, designing experimental studies to test the effectiveness of autonomy support training programs for teachers on student outcomes could be beneficial.
Finally, various effective constructs can be assessed for criterion validity, using scores on motivational orientations, self-regulation, self-image, and similar criteria.
Q8: It is suggested that the validated instrument be included as an annex to the article.
R8: Thank you for this suggestion, which will increase attention and citations.
|
Multi-Dimensional Perceived Autonomy Support Scale for Physical Education Beden Eğitimi için Algılanan Çok Boyutlu Özerklik Desteği Ölçeği |
|
|
|
|
|
My PE teacher understands my needs Beden eğitimi öğretmenim neye ihtiyacım olduğunu bilir. |
|
|
|
|
|
My PE teacher conveys confidence in my ability to do well in the lesson Beden Eğitimi Öğretmenim derste başarılı olacağıma inanır. |
|
|
|
|
|
My PE teacher allows me to express my opinion Beden eğitimi öğretmenim düşüncelerimi söylememe izin verir. |
|
|
|
|
|
My PE teacher is interested in what students want to do Beden Eğitimi Öğretmenim öğrencilerin yapmak istediği şeylerle ilgilenir. |
|
|
|
|
|
My PE teacher answers to me when I express my opinion Beden eğitimi öğretmenim düşüncelerimi ifade ettiğimde bana karşılık verir. |
|
|
|
|
|
My PE teacher explains why we learn certain exercises Beden eğitimi öğretmenim yapılan egzersizleri neden öğrendiğimizi açıklar. |
|
|
|
|
|
My PE teacher guides students in finding solutions Beden Eğitimi Öğretmenim öğrencilere çözüm bulma konusunda rehberlik eder. |
|
|
|
|
|
My PE teacher explains the effect of exercises Beden eğitimi öğretmenim yapılan çalışmaların etkilerini açıklar. |
|
|
|
|
|
My PE teacher offers hints on how to do better Beden eğitimi öğretmenim daha iyisini nasıl yapacağımızla ilgili ipuçları verir. |
|
|
|
|
|
My PE teacher gives an overview of a lesson at the beginning Beden eğitimi öğretmenim dersin başında dersin içeriğiyle ilgili genel bir açıklama yapar. |
|
|
|
|
|
My PE teacher allows me to do exercises using different methods Beden eğitimi öğretmenim yapılan egzersizlerde farklı yöntemler uygulamama izin verir. |
|
|
|
|
|
My PE teacher accepts different solutions in learning of exercise Beden eğitimi öğretmenim, alıştırmaları öğrenirken sunduğumuz farklı çözüm önerilerini kabul eder. |
|
|
|
|
|
My PE teacher allows me to choose between different exercises Beden eğitimi öğretmenim farklı egzersizler arasından seçim yapmama izin verir. |
|
|
|
|
|
My PE teacher allows me to choose an exercise place Beden eğitimi öğretmenim egzersiz yapacağım yeri seçmeme izin verir. |
|
|
|
|
|
My PE teacher allows me to choose sports equipment Beden eğitimi öğretmenim spor malzemesini seçmeme izin verir. |
|
|
|
Dear reviewer, thank you very much for your observations, criticism and comment.
Please see attachment.

Round 2
Reviewer 4 Report
Comments and Suggestions for Authors
Article: Adaptation of the Multidimensional Perceived Autonomy Support Scale in Physical Education to Grades 7-11 in Turkish: A Validity and Reliability Study
Areas for Improvement:
1. The authors have incorporated the suggestions in the abstract.
2. In the introduction, the authors have included several citations, but if necessary, they could conduct a more exhaustive literature review from the last 10 years.
3. The authors have improved the methodology section, but it is necessary to include a description of the design and type of research. The first subsection, population and sample (inclusion and exclusion criteria), techniques and instruments, data collection procedures, and type of analysis, have been significantly incorporated.
4. In the results section, the description of the results obtained has been improved, allowing for a better understanding.
5. In the discussion section, recent studies relevant to the study object have been integrated, allowing for a comparison of the findings with the current literature and reinforcing the scientific relevance of the results.
6. In the conclusion, we suggest including a subsection on new lines of research.
8. We suggest including the validated instrument as an appendix to the article.
Author Response
Dear Reviewer.
We wishes to express our gratitude for your review of our work and for the invaluable feedback you have provided, as well as for the time and effort you have dedicated to this task. Your observations, comments, and suggestions have proven to be immensely beneficial in enhancing the quality of our article. We have thoroughly considered each of your suggestions and criticisms and have taken great care to revise the content of our article accordingly. We have colored our annotations in red to facilitate your ability to distinguish them within the annotations.
Best Regards.
Q1. The authors have incorporated the suggestions in the abstract.
R1: Thank you very much for this observation and evaluation.
Q2. In the introduction, the authors have included several citations, but if necessary, they could conduct a more exhaustive literature review from the last 10 years.
R2: Dear reviewer, thank you for this valuable suggestion. The subject matter is both timeless and contemporary. So, the references we have selected for an integrated flow cannot solely derive from recent years. We have chosen not to forego our previous references and have endeavoured to incorporate new references from recent years following your suggestions. In adding these, we aimed to ensure that the current narrative flow remains unaffected by including new sentences. Consequently, we have added the following references:
If you would like to see where this link is inserted, take a see at the line: 82
Ng Betsy, Wu Hong Liu (2024). “Exploring the relationships among perceived teacher’s autonomy support, motivational regulations, and social-emotional outcomes. international journal of instruction, 17(4), 99-116. https://doi.org/10.29333/iji.2024.1746a
For the location of these three references, see lines 115-116
Tian, L., & Shen, J. (2023). The effect of perceived teachers’ interpersonal behavior on students’ learning in physical education: a systematic review. Frontiers in psychology, 14, 1233556.
https://www.frontiersin.org/journals/psychology/articles/10.3389/fpsyg.2023.1233556/full
Brandisauskiene, A., Buksnyte-Marmiene, L., Cesnaviciene, J., & Jarasiunaite-Fedosejeva, G. (2023). The relationship between teacher’s autonomy-supportive behavior and learning strategies applied by students: The role of teacher support and equity. Sage Open, 13(2), 21582440231181384. https://journals.sagepub.com/doi/epub/10.1177/21582440231181384
Guo, Q., Samsudin, S., Yang, X., Gao, J., Ramlan, M. A., Abdullah, B., & Farizan, N. H. (2023). Relationship between Perceived Teacher Support and Student Engagement in Physical Education: A Systematic Review. Sustainability, 15(7), 6039. https://doi.org/10.3390/su15076039
https://www.mdpi.com/2071-1050/15/7/6039
If you would like to see where this link is inserted, take a see at the line: 124
Cronin, L. D., Allen, J., Mulvenna, C., & Russell, P. (2018). An investigation of the relationships between the teaching climate, students’ perceived life skills development and well-being within physical education. Physical Education and Sport Pedagogy, 23(2), 181–196. https://doi.org/10.1080/17408989.2017.1371684
https://www.tandfonline.com/doi/full/10.1080/17408989.2017.1371684#abstract
Q3. The authors have improved the methodology section, but it is necessary to include a description of the design and type of research. The first subsection, population and sample (inclusion and exclusion criteria), techniques and instruments, data collection procedures, and type of analysis, have been significantly incorporated.
R3: Thank you for your warning about the missing piece in this important section of our work. This will help improve the perception and understanding of what we are doing.
Design of research
This study employed a quantitative approach, predicated upon a cross-sectional screening model that focuses on statistical analysis to validate the model across diverse student demographics. Such studies examine outcomes across large populations, often investigating differences among them. They are conducted at a single point in time, in contrast to longitudinal studies. May see in lines: 257-263.
Q4. In the results section, the description of the results obtained has been improved, allowing for a better understanding.
R 4: Thank you very much for this observation and evaluation.
Q5. In the discussion section, recent studies relevant to the study object have been integrated, allowing for a comparison of the findings with the current literature and reinforcing the scientific relevance of the results.
R5: Thank you very much for this commentary for discussion section.
Q6. In the conclusion, we suggest including a subsection on new lines of research.
R6: Thank you very much for this observation and evaluation. In the previous process, we added several promising elements. However, we haven't introduced new ones to avoid overdoing it. In one section, we suggested solutions for new research based on the challenges we faced. In other words, we aimed to avoid repetition. Thise sentences is seen in lines 860-889.
Q8. We suggest including the validated instrument as an appendix to the article.
R8: Your suggestion is valuable for ensuring our work gets read and used as a reference. It also highlights the importance of visibility. Thank you! We are been added both as supplementary file and added it to the end of the references. In line 1225.
|
Multi-Dimensional Perceived Autonomy Support Scale for Physical Education Beden Eğitimi için Algılanan Çok Boyutlu Özerklik Desteği Ölçeği |
|
|
|
|
|
My PE teacher understands my needs Beden eğitimi öğretmenim neye ihtiyacım olduğunu bilir. |
|
|
|
|
|
My PE teacher conveys confidence in my ability to do well in the lesson Beden Eğitimi Öğretmenim derste başarılı olacağıma inanır. |
|
|
|
|
|
My PE teacher allows me to express my opinion Beden eğitimi öğretmenim düşüncelerimi söylememe izin verir. |
|
|
|
|
|
My PE teacher is interested in what students want to do Beden Eğitimi Öğretmenim öğrencilerin yapmak istediği şeylerle ilgilenir. |
|
|
|
|
|
My PE teacher answers to me when I express my opinion Beden eğitimi öğretmenim düşüncelerimi ifade ettiğimde bana karşılık verir. |
|
|
|
|
|
My PE teacher explains why we learn certain exercises Beden eğitimi öğretmenim yapılan egzersizleri neden öğrendiğimizi açıklar. |
|
|
|
|
|
My PE teacher guides students in finding solutions Beden Eğitimi Öğretmenim öğrencilere çözüm bulma konusunda rehberlik eder. |
|
|
|
|
|
My PE teacher explains the effect of exercises Beden eğitimi öğretmenim yapılan çalışmaların etkilerini açıklar. |
|
|
|
|
|
My PE teacher offers hints on how to do better Beden eğitimi öğretmenim daha iyisini nasıl yapacağımızla ilgili ipuçları verir. |
|
|
|
|
|
My PE teacher gives an overview of a lesson at the beginning Beden eğitimi öğretmenim dersin başında dersin içeriğiyle ilgili genel bir açıklama yapar. |
|
|
|
|
|
My PE teacher allows me to do exercises using different methods Beden eğitimi öğretmenim yapılan egzersizlerde farklı yöntemler uygulamama izin verir. |
|
|
|
|
|
My PE teacher accepts different solutions in learning of exercise Beden eğitimi öğretmenim, alıştırmaları öğrenirken sunduğumuz farklı çözüm önerilerini kabul eder. |
|
|
|
|
|
My PE teacher allows me to choose between different exercises Beden eğitimi öğretmenim farklı egzersizler arasından seçim yapmama izin verir. |
|
|
|
|
|
My PE teacher allows me to choose an exercise place Beden eğitimi öğretmenim egzersiz yapacağım yeri seçmeme izin verir. |
|
|
|
|
|
My PE teacher allows me to choose sports equipment Beden eğitimi öğretmenim spor malzemesini seçmeme izin verir. |
|
|
|
